# ECD: A Machine Learning Benchmark for Predicting Enhanced-Precision Electronic Charge Density in Crystalline Inorganic Materials

**Pin Chen, Zexin Xu, Qing Mo, Hongjin Zhong, Fengyang Xu, Yutong Lu**
National Supercomputer Center in Guangzhou,
School of Computer Science and Engineering, Sun Yat-sen University
`{chenp85,xuzx25,moqing,luyutong}@mail.sysu.edu.cn`
`{zhonghj28,xufy9}@mail2.sysu.edu.cn`

## Abstract

Supervised machine learning techniques are increasingly being adopted to speed up electronic structure predictions, serving as alternatives to first-principles methods like Density Functional Theory (DFT). Although current DFT datasets mainly emphasize chemical properties and atomic forces, the precise prediction of electronic charge density is essential for accurately determining a system's total energy and ground state properties. In this study, we introduce a novel electronic charge density dataset named ECD, which encompasses 140,646 stable crystal geometries with medium-precision Perdew-Burke-Ernzerhof (PBE) functional data. Within this dataset, a subset of 7,147 geometries includes high-precision electronic charge density data calculated using the Heyd-Scuseria-Ernzerhof (HSE) functional in DFT. By designing various benchmark tasks for crystalline materials and emphasizing training with large-scale PBE data while fine-tuning with a smaller subset of high-precision HSE data, we demonstrate the efficacy of current machine learning models in predicting electronic charge densities. The ECD dataset and baseline models are open-sourced to support community efforts in developing new methodologies and accelerating materials design and applications.

## 1 Introduction

The electronic charge density (ECD) is a fundamental yet informative observable in materials physics. Density functional theory (DFT) demonstrates that the properties of materials in their ground state can be completely and uniquely determined by the ECD Lewis et al. (2021); Fabrizio et al. (2019). This theory has been widely applied across various physical systems, from individual molecules to crystalline solids, significantly advancing our understanding and control of the natural world. For example, bonding characteristics between neighboring atoms (covalent, ionic, metallic bonds) can be fully described through the ECD. Leveraging this property, novel materials with specific structures and target properties can be artificially designed Macchi (2013). Additionally, according to modern band structure theory, a wide range of electronic, magnetic, and optical properties, as well as their couplings-such as electrostatic moments, potentials and interaction energies, spin susceptibility, light absorption, and electromagnetic responses-can be directly derived from the ECD Kolb et al. (2017); Tan et al. (2021). Due to these advantages, obtaining the ECD has broad applications.

Obtaining ECD from experimental techniques, such as high-resolution electron diffraction Schmøkel et al. (2014); Chopra (2012), is time-consuming and complex. Theoretically, calculating ECD through DFT calculations is more convenient. However, this approach is computationally intensive due to its complexity of $O(n^3 T)$, where n is the number of electrons and T is the number of iterations in the structural optimization process Kohn & Sham (1965). Recently, machine learning methods have shown great potential in accelerating DFT computations. For example, various invariant geometric deep learning methods have been developed for representation learn-

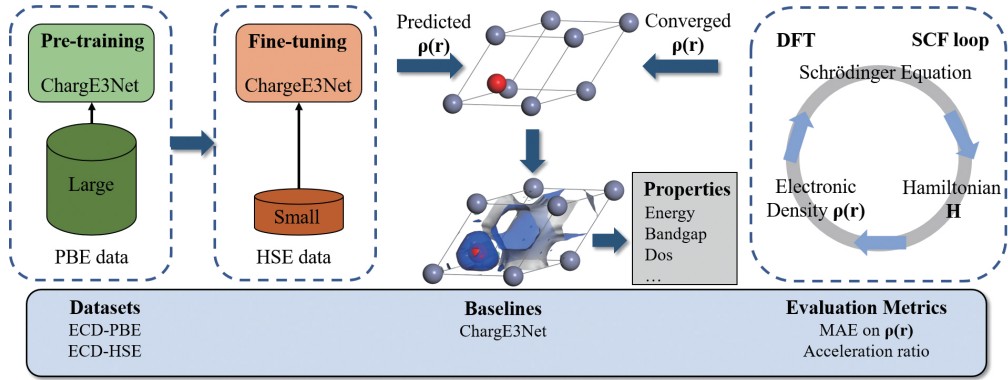

Figure 1: The objectives and scope of the proposed ECD dataset and benchmark.

ing of material and molecular structures, providing data-driven approximations to DFT calculations for faster predictions of physical and chemical properties Choudhary & DeCost (2021); Yan et al. (2022); Chen et al. (2022). Additionally, extensive DFT datasets have been generated for training machine learning models Jain et al. (2013); Choudhary et al. (2018); Kirklin et al. (2015). Notably, predicting ECD differs significantly from general physical and chemical property predictions, as the representation of charge density can be uniquely determined by three vectors and a scalar matrix in real or reciprocal space. In data-driven ECD prediction tasks, maintaining the equivariance of the quantum tensor network architecture is crucial for ensuring that the network can make physically meaningful predictions. Specifically, this equivariance can be represented by the rotational Wigner D-matrix, which may include higher-order rotations beyond three-dimensional space. Therefore, to conduct systematic and in-depth research on these tasks, it is necessary to generate large-scale ECD datasets.

To provide a realistic dataset and enable thorough evaluation, we meticulously constructed a charge density dataset named ECD. This dataset encompasses 140,646 stable crystal geometries calculated using the medium-precision Perdew-Burke-Ernzerhof (PBE) functional. Additionally, we carefully selected a subset of 7,147 geometries for high-precision electronic charge density calculations using the Heyd-Scuseria-Ernzerhof (HSE) functional. The construction of this large-scale dataset was a significant challenge due to the extensive computational resources required and the careful planning involved. To the best of our knowledge, this dataset represents the largest publicly available collection of DFT calculations that includes both PBE and HSE functionals. The creation of the ECD dataset demanded approximately 130 million CPU core hours on cutting-edge supercomputers. Notably, the HSE functional calculations alone consumed around 25 million core hours, accounting for 19.2% of the total computational resources. This substantial investment of computational time underscores the complexity and resource-intensive nature of generating high-precision charge density data at this scale. To conduct comprehensive studies on quantum tensor networks, we have designed three types of experiments: 1. Developing and evaluating methodologies for training models capable of accurate charge density prediction, specifically by training on large-scale PBE data and fine-tuning with a smaller subset of high-precision HSE data; 2. Investigating the impact of charge density prediction accuracy on related applications; 3. Assessing model performance on out-of-distribution (OOD) data to evaluate their suitability for real-world applications. To demonstrate the quality of the predicted charge density, we employ two metrics: MAE on charge density to evaluate the accuracy of the model's predictions, and acceleration ratio in DFT calculation to assess whether the predicted charge density can effectively speed up DFT computations. We present the ECD dataset, along with the benchmark objectives and scope, in Fig. 1.

## 2 BACKGROUND AND RELATED WORKS

### 2.1 DENSITY FUNCTIONAL THEORY

**DFT functional.** DFT functionals are mathematical expressions that describe the exchange-correlation energy, which is a crucial component in the total energy calculation. These functionals

vary in complexity and computational cost, typically following a Jacob's Ladder hierarchy Tran et al. (2016). As one ascends this ladder, functionals incorporate more physical effects and offer higher accuracy but at the expense of increased computational effort as shown in Figure 2. In this study, we focus on the PBE Perdew et al. (1998) and HSE functionals Heyd & Scuseria (2004). The PBE functional strikes a balance between accuracy and computational efficiency, making it highly cost-effective and widely used for constructing large-scale DFT computation databases. On the other hand, the HSE functional, although computationally intensive, offers higher accuracy in electronic structure information.

Specially, the PBE functional is a widely used exchange-correlation functional within the DFT framework. It belongs to the Generalized Gradient Approximation (GGA) Perdew et al. (1996) class of functionals, which improve upon the Local Density Approximation (LDA) Jackson & Pederson (1990) by including the gradient of the electron density. The PBE functional is designed to balance accuracy and computational efficiency for various systems.

The exchange-correlation energy $E_{xc}$ in the PBE functional is given by:

$$E_{xc}^{\text{PBE}} = \int \left( \epsilon_x^{\text{PBE}}(n(\mathbf{r}), \nabla n(\mathbf{r})) + \epsilon_c^{\text{PBE}}(n(\mathbf{r}), \nabla n(\mathbf{r})) \right) n(\mathbf{r}) \, d\mathbf{r} \qquad (1)$$

where $n(\mathbf{r})$ is the electron density at position $\mathbf{r}$, $\epsilon_x^{\text{PBE}}$ is the exchange energy density, and $\epsilon_c^{\text{PBE}}$ is the correlation energy density.

The HSE functional is a hybrid functional that incorporates a portion of exact exchange from Hartree-Fock theory Jiménez-Hoyos et al. (2012) with the PBE exchange-correlation functional. This combination enhances the accuracy for electronic structure calculations, especially for systems with localized d and f electrons.

The HSE functional modifies the PBE functional by including a screened Coulomb potential, which limits the range of the Hartree-Fock exchange interaction, thus improving computational efficiency. The HSE exchange-correlation energy is given by:

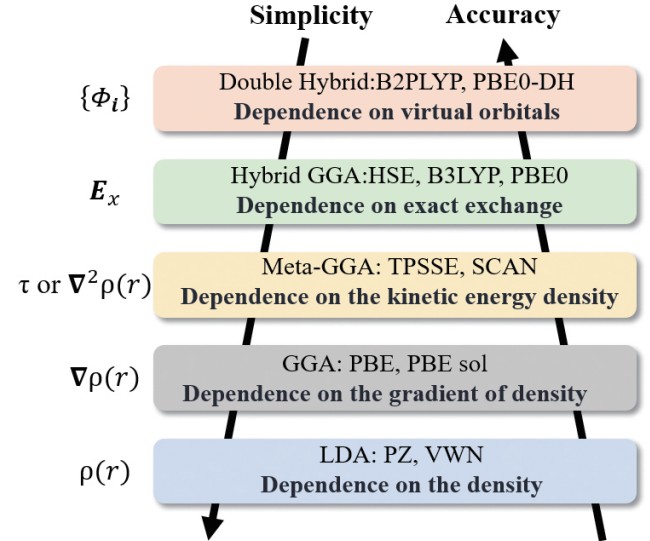

Figure 2: Jacob's ladder of density functional approximations. Higher rungs yield greater accuracy but more complexity.

$$E_{xc}^{\text{HSE}} = \alpha E_x^{\text{HF,SR}}(\omega) + (1 - \alpha) E_x^{\text{PBE,SR}}(\omega) + E_x^{\text{PBE,LR}}(\omega) + E_c^{\text{PBE}} \qquad (2)$$

where $E_x^{\text{HF,SR}}(\omega)$ represents the short-range Hartree-Fock exchange energy, $E_x^{\text{PBE,SR}}(\omega)$ and $E_x^{\text{PBE,LR}}(\omega)$ denote the short-range and long-range PBE exchange energies, respectively, and $E_c^{\text{PBE}}$ is the PBE correlation energy. The parameter $\alpha$ is the mixing parameter (typically around 0.25), and $\omega$ is the screening parameter.

The combination of these terms allows the HSE functional to achieve a better balance between accuracy and computational cost, making it suitable for a wide range of materials and electronic properties.

**Electronic Charge Density.** In DFT, the ECD $\rho(r)$ and the wave function $\Psi(r_1, r_2, \ldots, r_N)$ are related by:

$$\rho(r) = N \int d^3 r_2 \cdots \int d^3 r_N |\Psi(r, r_2, \ldots, r_N)|^2 \qquad (3)$$

The total number of electrons $N$ within a unit cell equals the integral of $\rho(r)$ over the entire volume $V_{\text{un}}$:

$$N = \int_{V_{\text{un}}} \rho(r)\, dV \tag{4}$$

After discretization, this relationship is expressed as:

$$N = \sum_{i=1}^{NGXF \cdot NGYF \cdot NGZF} \rho(r_i) \cdot \frac{V_{\text{un}}}{NGXF \cdot NGYF \cdot NGZF} \tag{5}$$

where $NG(X, Y, Z)F$ are the fine Fast Fourier Transform (FFT) grids in reciprocal space along the x, y, and z directions. The discrete values $\rho(r_i)$ at each fine FFT grid point are recorded in the charge density file, which contains all the necessary information about $\rho(r)$.

In non-spin-polarized calculations for non-magnetic materials, the ECD includes the total electronic charge density $\rho(r) = \rho_{\text{spin-up}}(r) + \rho_{\text{spin-down}}(r)$. For spin-polarized calculations of magnetic materials, an additional spin electronic charge density $\rho_{\text{spin}}(r) = \rho_{\text{spin-up}}(r) - \rho_{\text{spin-down}}(r)$ is also provided.

## 2.2 Higher-order Equivariant matrices

Given a crystal material structure $M = (A, X, L)$ where $A$ represents atomic species, $X$ denotes atomic positions, and $L$ is the unit cell, we aim to predict the electronic charge density $\rho(\vec{r})$ : $\mathbb{R}^3 \rightarrow \mathbb{R}$ at any point $X$. Similar to other tensor properties such as Hamiltonian and forces, the charge density preserves the $SE(3)$ equivariance of the atomic system. This means that rotations and translations of the atomic system in Euclidean space result in equivalent transformations of the charge density. Representing graph structures using only invariant scalar features (e.g., interatomic distances) can achieve $SE(3)$ invariance, but lacks angular information, thus limiting the model's accuracy. Moreover, incorporating vector $\mathbb{R}^3$ features to integrate angular information has been shown to enhance the performance of charge density predictions Jørgensen & Bhowmik (2022).

Translation invariance is maintained by utilizing relative atomic coordinates, while rotation equivariance is ensured by confining features to the irreducible representations (irreps) of SO(3), which are manipulated by equivariant functions. These features are denoted as $V_{cm}^{(\ell,p)}$, a collection of tensors indexed by rotation order $\ell \in \{0, 1, 2, \ldots\}$ and parity $p \in \{-1, 1\}$. Each tensor has a channel index $c \in [0, N_{\text{channels}})$ and an index $m \in [-\ell, \ell]$. Consequently, the representation for specific $\ell$ and $p$ dimensions is of size $\mathbb{R}^{N_{\text{channels}} \times (2\ell+1)}$. These representations are combined using the equivariant tensor product $\otimes$ with Clebsch-Gordan coefficients $C$ as detailed in Thomas et al. (2018) and implemented in e3nn Geiger & Smidt (2022):

$$\left(U_{(\ell_1, p_1)} \otimes V_{(\ell_2, p_2)}\right)_{cmo}^{(\ell_o, p_o)} = \sum_{m_1=-\ell_1}^{\ell_1} \sum_{m_2=-\ell_2}^{\ell_2} C_{(\ell_o, m_o)}^{(\ell_1, m_1)(\ell_2, m_2)} U_{cm_1}^{(\ell_1, p_1)} V_{cm_2}^{(\ell_2, p_2)} \tag{6}$$

where $\ell_o$ and $p_o$ are defined by $|\ell_1 - \ell_2| \leq \ell_o \leq |\ell_1 + \ell_2|$ and $p_o = p_1 p_2$. We retain only those representations with $\ell_o \leq L$, where $L$ is the maximum allowed rotation order.

## 2.3 Datasets for Crystalline Inorganic Materials

Materials databases such as JARVIS Choudhary et al. (2018), OQMD Kirklin et al. (2015), NOMAD Draxl & Scheffler (2019), Materials Project (MP) Jain et al. (2013), and AFLOW Curtarolo et al. (2012) primarily provide data on properties such as energy, electronic structure, mechanical, and magnetic properties. However, only NOMAD and MP provide the option to download charge density files. The charge density files in NOMAD are generated during structure optimization calculations, not from the self-consistent process, and the parameters used for these calculations are missing, making reliable subsequent processing from these data challenging. MP provides approximately 12,000 electronic structure datasets, accessible via the officially released API. However, the charge density in MP is calculated using the PBE functional, which has limitations for electronic structure calculations Saßnick & Cocchi (2021), especially when dealing with strongly correlated

materials. Additionally, there are smaller databases focused on charge density data, such as the ECD-cubic database Wang et al. (2022) with 17,418 entries, but it is limited to cubic inorganic materials, restricting its applicability. MaterialGo Jie et al. (2019) is the only reported database containing electronic structure data from approximately 10,000 HSE functional calculations. However, its query and download services are currently unavailable. Additionally, since this database uses the PWmat software package Jia et al. (2013), combining its data with existing PBE data calculated using other software might introduce biases due to differences in the software and computational parameters. In this study, we constructed a dataset of nearly 140,000 entries with PBE precision and a subset of approximately 7,000 entries with HSE precision by utilizing supercomputing power. Apart from the specific calculation parameters of these two functionals, all other aspects, such as the software and computational environment, remain consistent. By leveraging this mixed-precision dataset, we aim to develop an electronic charge density prediction model with improved accuracy.

## 2.4 DATASETS

**Dataset Generation**. All materials structures are obtained from the Matgen database Chen et al. (2022). We perform DFT computations using the Vienna Ab initio Simulation Package (VASP) version 5.4.4 Kresse & Furthmüller (1996); Hafner (2008). All systems are fully relaxed with respect to volume and atomic coordinates using the GGA Perdew et al. (1996) of PBE, with pseudopotentials from the projector augmented wave (PAW) method Kresse & Joubert (1999) at zero temperature and pressure. A unified plane wave cutoff of 520 eV is used for all structures to ensure consistency and compatibility. This cutoff energy also satisfies the condition of being 1.25 times the maximum energy cutoff of the element plane wave basis sets in the pseudopotentials used. Due to the limitations of pure GGA in accurately describing electronic interactions in strongly correlated materials containing transition metals or rare earth elements with 3d/4f orbitals, we employ the GGA+U method, with all U values are same as Chen et al. (2022). Brillouin-zone integrations are performed using the $\Gamma$-centered Monkhorst-Pack scheme. We begin with a k-point mesh with a dense sampling mesh of $2\pi \times 0.02^{-1}$. For a small number of structures that are difficult to converge, we use a medium-dense sampling mesh of $2\pi \times 0.04^{-1}$. The blocked Davidson iteration scheme is used to solve the Kohn-Sham (KS) equations, with convergence criteria set to 0.1 meV for energy and 0.001 eV/ for force.

For the DFT calculations with the HSE functional (HSE06), hybrid functional calculations are enabled, with the exact exchange mixing parameter set to 0.25 and the screening parameter set to 0.2. Additionally, the time step for the calculation is specified with 0.4. These settings ensure that the HSE functional provides more accurate electronic structure information with appropriate adjustments for exact exchange and screening. Considering that calculations using the HSE functional are over ten times slower than those using the PBE functional, we selected a subset of 7,147 structures from an initial pool of 140,646 for computation. The selection criteria is as follows: structures with fewer than 20 atoms, a diverse range of element types encompassing all elements, inclusion of unary, binary, and ternary compositions, and representation across various crystal systems. Through this selection strategy, the HSE functional data domain is ensured to closely approximate the PBE functional data space. More statistical information is presented in the Appendix A.

We utilized approximately 1,000 computing nodes (64 cores per node) on the supercomputer for large-scale high-throughput computations. In total, approximately 130 million CPU core hours were utilized, with the HSE functional alone consuming around 25 million core hours, constituting 19.2% of the overall computational resources. The DFT calculation workflow is available at: https://anonymous.4open.science/r/DFTflow-635A. Our benchmark is publicly available at: https://anonymous.4open.science/r/ECDBench-037F.

**Dataset Statistics**. The statistical data, including the element distribution and structure size histogram for ECD, is presented in Figure 2. These material structures include 94 different elements, covering nearly all commonly found elements in the periodic table. Existing datasets like MP and OQMD did not feature the elements Cm, Rn, Ra, Po, and Am. In terms of material size (number of atoms per crystal cell), about half of the structures in ECD have 20 or fewer atoms, and the dataset includes approximately 9,000 large structures with over 80 atoms per unit cell. Specifically, we selected 7,147 structures for HSE functional DFT calculations. The selection strategy aimed to maximize the resemblance of elemental distribution to the 140,646 dataset. To further reduce com-

putational time, we focused on unary, binary, and ternary materials with fewer than 20 atoms per unit cell. More detailed information is provided in Appendix B.

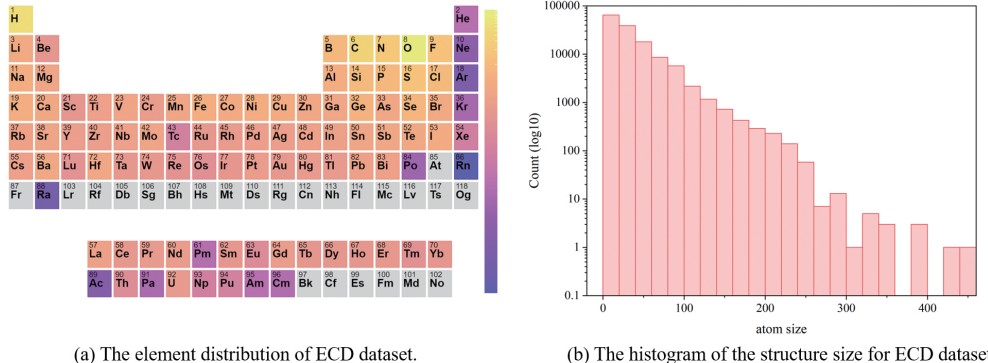

(a) The element distribution of ECD dataset.  (b) The histogram of the structure size for ECD dataset.

Figure 3: The dataset statics for ECD dataset with 140, 646 entries, including element distribution and structure size histogram.

## 2.5 TASKS

To thoroughly evaluate the performance of charge density predictions, we define the following tasks utilizing PBE functional and HSE functional precision from the ECD dataset, and the MP PBE precision pretrained model Koker et al. (2024). The statistics of our defined tasks is presented in Table 1.

Table 1: The statistics of our defined four tasks.

| Task | Total entries | Traning/validation/testing entries |
|---|---|---|
| ECD-PBE-id | 140,646 | 138,134/512/2000 |
| ECD-MP-id | 2000 | -/-/2000 |
| ECD-HSE-id | 5,647 | 4147/500/1000 |
| ECD-PBE_HSE-id | 1000 | -/-/1000 |
| ECD-PBE_HSE_tuning-id | 7147 | 5647/500/1000 |
| EXP-id | 41 | -/-/41 |
| OOD-id | 2000 | -/-/2000 |

**ECD-PBE-id**. We partition the PBE data in the ECD dataset following the method Koker et al. (2024): 138,134 for training, 512 for validation, and 2000 for testing. This forms the primary evaluation task for predicting charge density matrices.

**ECD-MP-id**. In this task, we use the model from ECD-PBE-id task to predict the test data of the ChargE3Net model in the MP dataset for direct comparison with existing models Koker et al. (2024). We perform a similarity analysis between the test data in the MP dataset and the training data in the ECD dataset, ensuring no identical data points are present in the training set. This task is designed to enable a fair comparison with existing charge density models.

**ECD-HSE-id**. In this task, the dataset comprises 7,147 HSE-calculated data points, which are partitioned into 5,647 for training, 500 for validation, and 1,000 for testing. This task is designed to train a model solely on HSE data, serving as a baseline for comparison with a model pretrained on PBE data.

**ECD-PBE_HSE-id**. We divide the HSE dataset into training, validation, and test sets with 5647, 500 and 1000. We then used the model from the ECD-PBE-id task to predict the HSE test data to determine the gap between PBE functional and HSE functional. This task aims to assess whether a model trained on PBE data can accurately predict HSE data directly.

**ECD-PBE_HSE_tuning-id**. We fine-tune the model from the ECD-PBE-id task on the HSE dataset. The trained model is then tested on HSE test set (as in ECD-PBE_HSE-id). By comparing the results with those of ECD-PBE_HSE-id, we determine whether the fine-tuned model achieved improved accuracy.

**EXP-id**. Building upon the work of Chen et al. (2022), we have compiled a dataset consisting of 47 entries that encompass material properties from wet-lab experiments, alongside various machine learning predictions, as well as PBE and HSE computational data. This dataset is intended to assess the potential of electronic charge density prediction models for real-world applications.

**OOD-id**. To further evaluate the model's generalization capability in real-world applications, we selected 2,000 material structures from the GNoME dataset and conducted DFT calculations to obtain their electronic charge densities. The structures in the GNoME dataset exhibit novelty and possess a distribution distinct from our existing database, making them suitable for assessing the model's out-of-distribution (OOD) performance.

## 2.6 METHODS

Machine learning models have been developed to address the challenges of predicting charge density, relying on custom-fitted basis functions. Early approaches utilized symmetry-adapted Gaussian process regression to predict coefficients for atom-centered basis functions, expressing the structural similarity and geometric relationships between target and training atomic environments through kernel functions Fabrizio et al. (2019); Grisafi et al. (2018). Recent work has employed invariant and equivariant neural networks to predict these coefficients from atomic features, using characteristics computed from density-functional tight-binding calculations as inputs Qiao et al. (2022). High-order equivariant neural networks have been used to predict coefficients of small molecules found in self-consistent DFT calculations and directly from atomic features Rackers et al. (2023). Despite achieving high accuracy in specific cases, these methods are limited by the expressivity of the density fitting basis sets. These atom-centered basis sets, typically defined per species, are numerically more challenging to converge for solid-state systems than plane-wave basis sets, restricting their applicability to molecular systems in vacuum.

As an alternative, several methods have been proposed to learn electron density directly from a discretized grid of density points. Charge density predictions on grids are agnostic to the basis set used for ground-truth quantum chemistry calculations and serve as a natural input format for plane-wave DFT codes. By inserting each grid or "probe" point into the atomic graph, charge density prediction can be modeled as a node regression task Cho et al. (2021); Sunshine et al. (2023). However, these models mainly focus on small and specialized atomic systems. Invariant graph convolutional networks have been trained on small datasets of crystalline polymers, demonstrating transferability to unseen structures. Invariant graph convolutional networks Schütt et al. (2018); Choudhary & DeCost (2021), have shown fast and accurate charge density prediction in small molecules, lithium-ion battery cathode materials, and electrolytes. Improvements in accuracy have been demonstrated using equivariant graph convolutional networks like PaiNN Schütt et al. (2021), which leverage $\mathbb{R}^3$ vector representations and rotationally equivariant operations. While these models perform well on small, specialized datasets, their expressive power on larger, diverse datasets like the MP dataset is not yet fully understood. ChargE3Net Koker et al. (2024) constructs rotationally equivariant networks using vector representations and improves model accuracy by incorporating higher-order equivariant features, showing diverse performance on the MP dataset. Therefore, the equivariant quantum tensor network ChargE3Net is selected as the primary benchmark for the ECD dataset.

## 2.7 METRICS

To assess the quality of the predicted charge density, we utilize several metrics that evaluate both approximation precision and computational performance.

**MAE on $\rho(r)$.** We evaluate our models' performance using the mean absolute error (MAE) normalized by the total number of electrons within the atomic system's volume $\epsilon_{\text{mae}}$ Koker et al. (2024), calculated via numerical integration on the charge density grid points $G$:

$$\epsilon_{\text{mae}} = \frac{\sum_{\vec{g}_k \in G} |\rho(\vec{g}_k) - \hat{\rho}(\vec{g}_k)|}{\sum_{\vec{g}_k \in G} |\rho(\vec{g}_k)|} \tag{7}$$

Unless otherwise stated, the probe points $G$ refer to the complete set of discretized unit cell grid points for which DFT-computed charge density values are available.

**Acceleration Ratios.** we use achieved ratio and error-level ratio to assess the effectiveness of the predicted charge density $\rho(r)$ in expediting DFT calculations. The achieved ratio measures the number of optimization steps required when initializing with the predicted charge density compared to traditional initial guess methods. A well-predicted charge density enables the Self-Consistent Field (SCF) algorithm to converge more quickly, significantly reducing the number of optimization steps. Conversely, the optimal ratio evaluates the number of single optimization step for each material relative to the total number of steps, serving as a benchmark for ideal performance.

## 3 EXPERIMENTS

### 3.1 SETUP

To assess the performance of deep learning approaches on the proposed dataset, we carry out experiments on the four designed tasks as described in Section 2.5. Specifically, we evaluate the performance of ChargE3Net, a network specifically designed for efficient and accurate prediction of electronic charge density. ChargE3Net is known for its effectiveness and efficiency in handling the task at hand, making it a suitable testing method for our benchmark evaluation. For quantitative evaluation, we use the metrics introduced in Section 2.7. We train the models using an Nvidia A800 GPU and an Intel Xeon Gold 6348 CPU.

Following the model setup in ChargE3Net, we employ four node-wise interaction layers to aggregate messages from neighboring nodes and update the node irreducible representations in all implemented models. We train all models with a total training step of 1,000,000 using a batch size of 16 with 200 charge density probe points per batch. For each gradient step, a random batch of materials is selected, from which a random subset of the charge density probe points is used. To expedite the convergence of model training, we implement a learning rate scheduler. The scheduler starts with a learning rate of 0.005, which is decayed by $0.96^{s/\beta}$ at step s, where $\beta$ is set to 3 x $10^3$. We use L1 error as the loss function for optimization. The Adam optimizer Kingma & Ba (2014) is used for training the models. During the model tuning phase, we fine-tune our model on the ECD-PBE-id task, allowing all parameters to be adjusted during training.

### 3.2 RESULTS AND DISCUSSION.

**Overall performance on ECD dataset.** Initially, we assess the model's comprehensive performance across the designated tasks by examining the accuracy of the predicted charge density matrices on the test set. As illustrated in Table 2, the ChargE3Net models employed exhibit a satisfactorily low MAE in predicting the charge density matrices for all specified tasks. We observe the following: 1. ChargE3Net demonstrates strong performance on the MP dataset and shows further improvement on our ECD dataset, indicating that an increase in data scale contributes to superior performance; 2. Electronic charge density is an equivariant property, which explains why equivariant models achieve optimal performance across both the MP and ECD datasets; 3. Training on electronic charge density data from the PBE functional, followed by fine-tuning with HSE functional data, helps mitigate the limitations of each dataset in terms of accuracy and data scale.

We evaluate the efficacy of the predicted charge density by examining its impact on accelerating DFT calculations. As outlined in Section 2.7, we compare the number of optimization steps required when starting with the predicted charge density to those needed with conventional initial guess methods Lehtola (2019). Using VASP for DFT calculations, we determine the average optimization step ratio for 19 randomly selected materials in each dataset. Table 3 presents the ratio metrics,

Table 2: The overall performance on the testing set on the defined tasks.

| Dataset | Model | MAE (eV) |
|---|---|---|
| MP-PBE | invDeepDFT | 1.293±0.03 Jørgensen & Bhowmik (2022) |
| MP-PBE | DeepDFT | 1.212±0.02 Jørgensen & Bhowmik (2022) |
| MP-PBE | ChargeE3Net | 0.523±0.01 Koker et al. (2024) |
| ECD-PBE_MP-id | ChargeE3Net | **0.520±0.01** |
| ECD-PBE-id | ChargeE3Net | **0.685±0.03** |
| ECD-PBE-id | invChargeE3Net | 0.732±0.02 |
| ECD-HSE-id | ChargeE3Net | 1.534±0.07 |
| ECD-PBE_HSE-id | ChargeE3Net | 2.156±0.08 |
| ECD-PBE_HSE_tuning-id | ChargeE3Net | **1.085±0.03** |

showing the number of optimization steps needed when initialized with the model prediction relative to traditional DFT initialization. The results indicate that starting from the predicted charge density matrices provided by ChargE3Net reduces the number of optimization steps required to reach the converged charge density, suggesting that the predicted charge density is close to the convergence condition. These findings illustrate the potential of machine learning methods in expediting DFT calculations.

Table 3: The performance of DFT calculation acceleration. Both models, trained on the ECD-PBE-id split and the ECD-PBE_HSE_tuning-id split respectively, are evaluated on 19 randomly selected materials from the intersection of their test sets.

| Training Dataset | Metric | Ratio |
|---|---|---|
| ECD-PBE-id | Optimal ratio | 0.061±0.016 |
| | Achieved ratio ↓ | 0.757±0.073 |
| ECD-PBE_HSE_tuning-id | Optimal ratio | 0.071±0.033 |
| | Achieved ratio ↓ | 0.681±0.113 |

We have conducted additional experiments as suggested to evaluate the sufficiency of the HSE dataset and its impact on model performance. The results are summarized in Table 4. As the data indicates, increasing the proportion of high-quality HSE data consistently improves the model's predictive accuracy and accelerates the computation. This trend suggests that further increasing the amount of HSE data could continue to enhance both accuracy and acceleration.

Table 4: Impact of HSE data ratio on MAE and acceleration rate.

| HSE Data Ratio (%) | MAE (eV) | Acceleration Rate |
|---|---|---|
| 0 | 2.156 ± 0.08 | 0.757 ± 0.073 |
| 25 | 2.057 ± 0.07 | 0.743 ± 0.081 |
| 50 | 1.524 ± 0.10 | 0.714 ± 0.082 |
| 75 | 1.203 ± 0.06 | 0.692 ± 0.098 |
| 100 | 1.085 ± 0.03 | 0.681 ± 0.113 |

**Evaluation on wet-experimental data.** To further demonstrate the potential of predicting electronic charge densities in real-world material applications, we conducted additional experiments utilizing 41 experimentally measured values to emphasize the importance of accurate ECD predictions. Specifically, we focused on the band gap, a critical property for applications such as semiconductors, solar cells, and lighting technologies. Additionally, we compared our results with other methods for determining band gaps, including machine learning approaches and DFT calculations using both the PBE and HSE functionals. The results, summarized in Table 5, highlight the value of using high-precision HSE data for improved band gap predictions, as supported by Chen et al. (2022). We can observe that the ECD-PBE_HSE model demonstrates a significant improvement in accuracy compared to direct GNN predictions, and it also outperforms CrystalNet-TL, a model utilizing

transfer learning with HSE functional data. However, this increased accuracy comes at the cost of approximately 10 times more computational time. Despite this, the charge density prediction approach offers substantial benefits over direct DFT calculations, as HSE functional calculations take approximately 10 times longer than PBE functional calculations. Therefore, using electronic charge density predictions strikes an advantageous balance between accuracy and computational efficiency, proving highly valuable for band gap prediction in materials science.

Table 5: Performance comparison of ML and ECD-Based models on wet-experimental data.

| Model/Dataset | MAE (eV) | Time per structure |
|---|---|---|
| **ML Models** | - | (s) |
| CGCNN | 1.45 Chen et al. (2019) | 1.5 |
| MEGNet | 1.36 Chen et al. (2019) | 1.34 |
| CrystalNet | 1.19 Chen et al. (2022) | 1.67 |
| CrystalNet-TL | 0.70 Chen et al. (2022) | 1.58 |
| **ECD-Based Models** | - | (min) |
| ECD-PBE | 1.17 | 14.4 |
| ECD-PBE_HSE | 0.65 | 14.4 |
| **PBE-Based Datasets** | - | (min) |
| MP | 1.38 Choudhary et al. (2018) | - |
| Matgen | 1.21 Chen et al. (2022) | 24.5 |
| AFLOW | 1.20 Choudhary et al. (2018) | - |
| OQMD | 1.09 Choudhary et al. (2018) | - |
| **HSE-Based Dataset** | - | (min) |
| HSE | 0.41 Choudhary et al. (2018) | 228.1 |

ECD prediction models are particularly adept at capturing material properties such as energy, forces, and band gaps in a more comprehensive manner, as demonstrated in the ChargE3Net study Koker et al. (2024). These models offer superior accuracy and generalization, especially in complex systems, when compared to direct property prediction methods. Furthermore, ECD-based models are capable of achieving chemical accuracy in non-self-consistent calculations, which enhances their reliability and expands their potential applications within materials science.

**OOD evaluation using GNoME dataset.** To further validate the model's ability to predict electronic charge densities in truly novel materials, we using GNoME dataset Merchant et al. (2023) to evaluation our module for real-world application. Specially, we selected 2,000 structures from the GNoME dataset and conducted DFT calculations using the PBE functional to generate the necessary charge density data. The OOD evaluation on this new dataset yielded an MAE of 0.643 eV, which is comparable to the model's performance on the MP dataset (0.523 eV) and the ECD dataset (0.685 eV) of same scale, thereby demonstrating the model's robust generalization capability. These findings support the robustness and broader applicability of models trained on the ECD dataset, further highlighting its value in the field.

## 4 CONCLUSION

In this study, we focus on accelerating the computation and prediction of electronic charge densities, which are critical for elucidating the electronic properties of materials and driving the design of novel materials. To this end, we introduce ECD, a novel and publicly accessible dataset comprising 140,646 stable crystal structures computed with the medium-precision PBE functional. Additionally, ECD includes a subset of 7,147 crystal structures with high-precision electronic charge density data calculated using the more accurate HSE functional. We have undertaken rigorous and comprehensive experiments to validate the integrity and applicability of the dataset, ensuring that it meets high standards of quality and relevance for real-world applications. The dataset is designed to support large-scale benchmarking and the training of machine learning models, demonstrating their capability in predicting electronic charge densities with remarkable accuracy. Furthermore, this dataset opens up new avenues for advancing computational materials science, by providing a robust foundation for the development of data-driven models that can accelerate the discovery and optimization of new materials with tailored properties.

ACKNOWLEDGMENTS

This research is supported by the National Natural Science Foundation of China (62461146204).

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

## A  LIMITATION

While this study introduces an extensive, mixed-precision dataset of electronic charge densities, further improvements in data quality remain essential, particularly through increasing the ratio of high-precision data. Our experimental findings demonstrate that expanding the dataset with additional HSE functional data can further enhance model accuracy; however, this improvement necessitates substantial computational resources. Moreover, due to the high dimensionality of charge density data, which often involves over 1,000 electrons, the complexity of handling node and edge interactions within graph neural networks presents significant computational challenges. This complexity underscores the need for more efficient algorithms and optimization strategies. Consequently, enhancing the efficiency of model training and extending the number of training epochs are critical areas for future research, especially in balancing the trade-off between computational cost and model performance. Additionally, addressing these limitations will enable the scalability of this approach to larger, more complex systems, thereby broadening the applicability of the dataset to a wider range of materials science problems.

## B  DATA GENERATION

The first part primarily consists of high-quality, literature-reported structures from sources such as ICSD[1] and COD Graulis et al. (2012). We have collected experimental crystal structures from the ICSD and COD databases, totaling 621,782 structures, and ultimately acquired 76,463 DFT-calculated data entries, as shown in Figure 4a.

The second part comprises structures generated through algorithmic methods, representing novel and hypothetical structures not previously observed. Using the DiffCSP Jiao et al. (2024) crystal structure generation method and ab initio algorithms, we construct a large-scale set of hypothetical crystal structures, which are further refined using the methods described below.

1. **Reasonable chemical compositions**. We remove number the structure with number of chemical components that is more than 10.
2. **Electroneutrality approach**: We calculate the oxidation states of each element using the SCMAT toolkit Davies et al. (2019) and remove structures with charge imbalance.
3. **Exhibiting symmetry**: Structures with space group equal to 1 will be excluded.

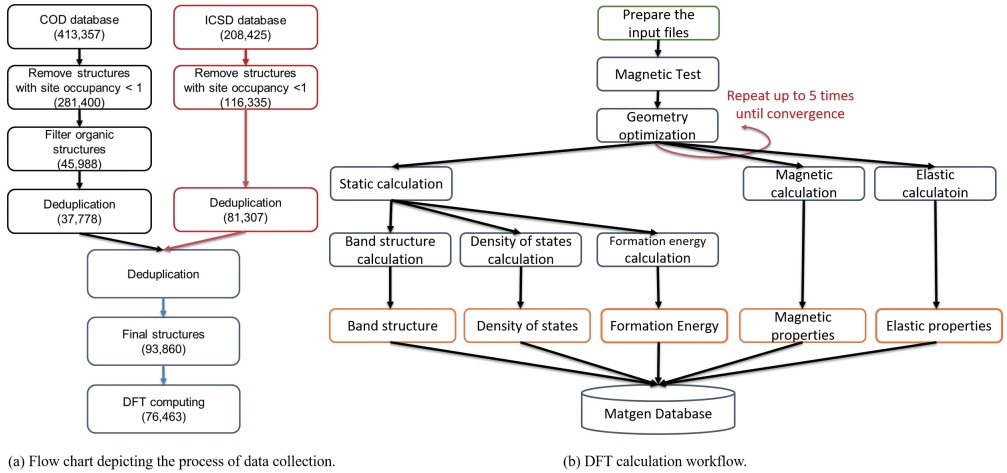

(a) Flow chart depicting the process of data collection.

(b) DFT calculation workflow.

Figure 4: The flowchart illustrating the data collection process and DFT calculation workflow.

Next, we employ DFT methods for structural optimization and self-consistent calculations on the selected results, as depicted in Figure 4b. All charge density files are obtained from the static calculation step within the workflow. To further obtain potentially stable structures, we select about

[1]https://icsd.products.fiz-karlsruhe.de

70,000 hypothetical crystal material calculation results based on formation energy. The distribution of formation energy is shown in Figure 5, ranging from -4 to 4 eV, which is consistent with existing databases such as MP Jain et al. (2013) and OQMD Kirklin et al. (2015).

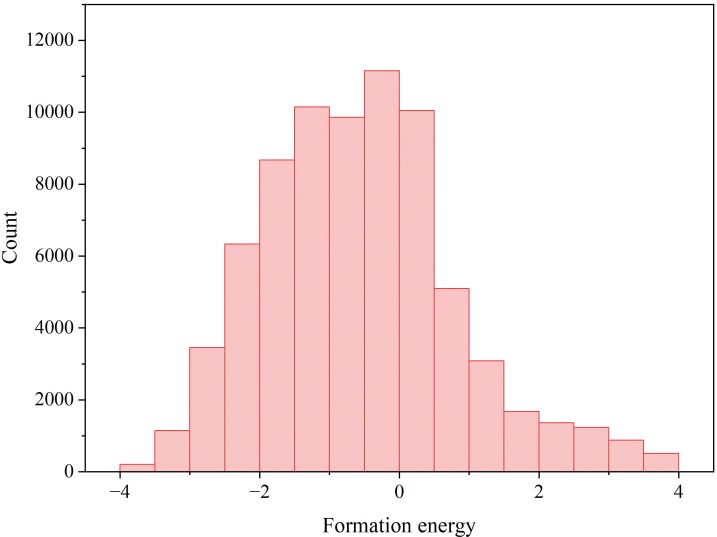

Figure 5: The distribution of formation energy.

## C    Data statistics

Figure 6 illustrates the statistical distribution of data from HSE functional DFT calculations. This dataset encompasses 94 elements (Figure 6a), consistent with the elemental distribution in PBE functional DFT data. Additionally, the dataset includes unary, binary, and ternary compounds, with ternary compounds comprising over half of the total (Figure 6b). Regarding the distribution of the number of atoms within the unit cells, all structures contain fewer than 20 atoms, with structures containing more than 10 atoms accounting for approximately one-third of the dataset (Figure 6c).

## D    Model Descriptions

In Table 2, the performance of ChargE3Net is compared with other models such as invDeepDFT, DeepDFT, and invChargE3Net. Below, we provide a detailed description of these models:

- **DeepDFT**: This model utilizes invariant graph neural networks (GNNs) where the edge features are represented by distances between vertices (atoms). It is designed to capture fundamental pairwise interactions in the system but does not incorporate directional information.

- **invChargE3Net**: We implemented this model by following the methodology of DeepDFT to ensure a fair comparison. In this invariant version, the edge features are represented solely by the distances between vertices, similar to DeepDFT, and the model does not incorporate directional or equivariant information.

- **EquiChargE3Net**: In the equivariant version of ChargE3Net, edge features include both distances and the directions of edges as features. This addition allows the model to capture directional dependencies in the data, which is essential for accurately modeling charge density and other physical quantities.

- **invDeepDFT**: This is an invariant version of the DeepDFT model where, like invChargE3Net, the edge features are limited to distances between vertices, ensuring that the model respects invariance properties but lacks directional information.

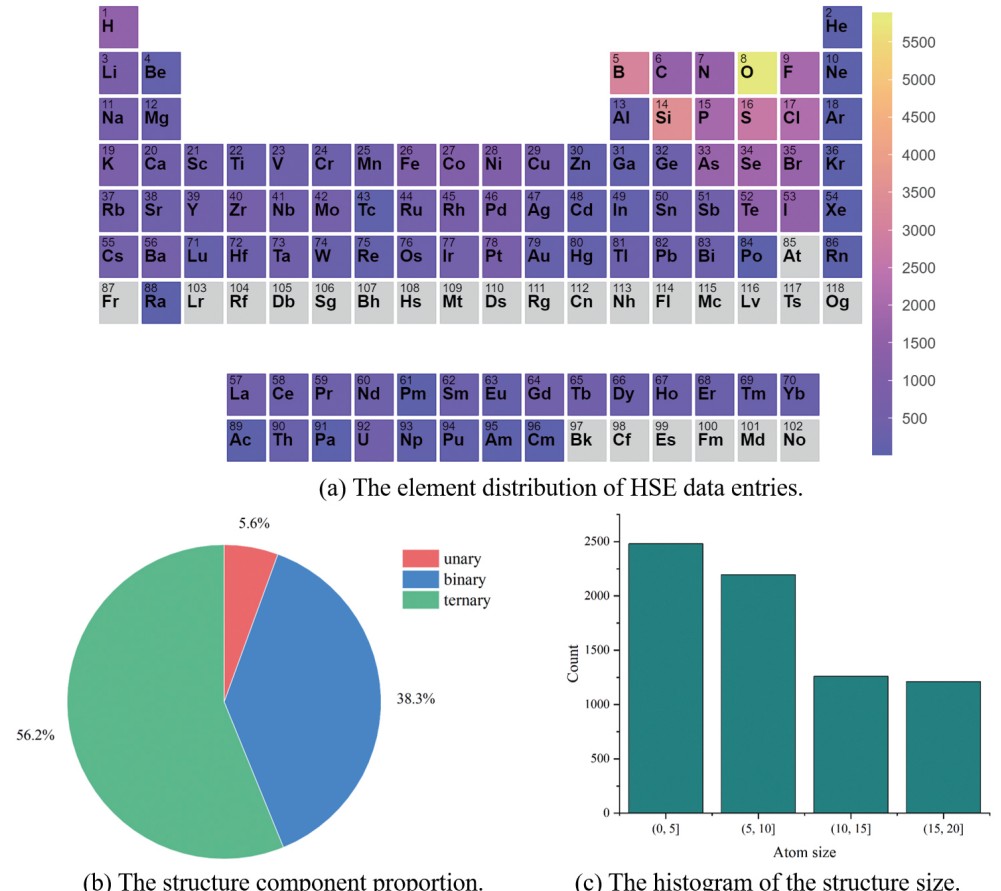

(a) The element distribution of HSE data entries.

(b) The structure component proportion.

(c) The histogram of the structure size.

Figure 6: The dataset statics for ECD dataset with 7147 HSE entries, including element distribution, structure component proportion and structure size histogram.

By incorporating directional features in EquiChargE3Net, the model achieves a higher degree of expressiveness compared to its invariant counterparts, such as DeepDFT and invChargE3Net, thereby enabling improved performance on tasks requiring the representation of directional dependencies.

## E   HSE CALCULATION DETAILS

We have provided the 41 experimental data points from the HSE-based dataset Choudhary et al. (2018) in the supplementary materials. These data were obtained through first-principles calculations using the HSE functional. Additionally, we have included a scatter plot comparing the calculated results with experimental values. It is worth noting KCl and CaO, typically exhibit larger errors. Moreover, HSE functional calculations are computationally expensive due to the complexity of solving hybrid functionals, which require a significantly greater number of iterations to converge. We also present the distribution of computation times for each data point. Materials with a higher number of atoms and elements located later in the periodic table, such as MoSe2 and ZrO2, exhibit the longest computation times.

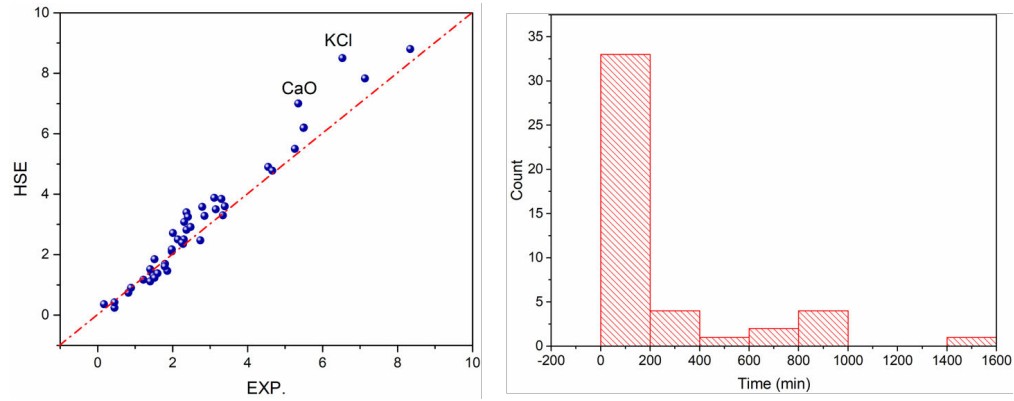

Figure 7: The performance characteristics of HSE calculation in wet-experimental evaluation.

## F   CASE STUDIES

we have included a failure case for further clarification and analysis (code: ebfc3a7854). We present a comparison plot of the predicted results and DFT results, where the visualization reveals that Charge3Net predicts the electron count of certain atoms, such as Be and Ce, to be inconsistent with the target atom.

## G   DATA FILE SYNTAX EXAMPLE

The general structure of the ECD file is as follows:

```
System name
Scaling factor
Lattice vector 1
Lattice vector 2
Lattice vector 3
Element names and atom counts
Coordinate system (Direct/Cartesian)
Atomic coordinates
FFT grid dimensions
Charge density data (flattened)
Augmentation occupancies
```

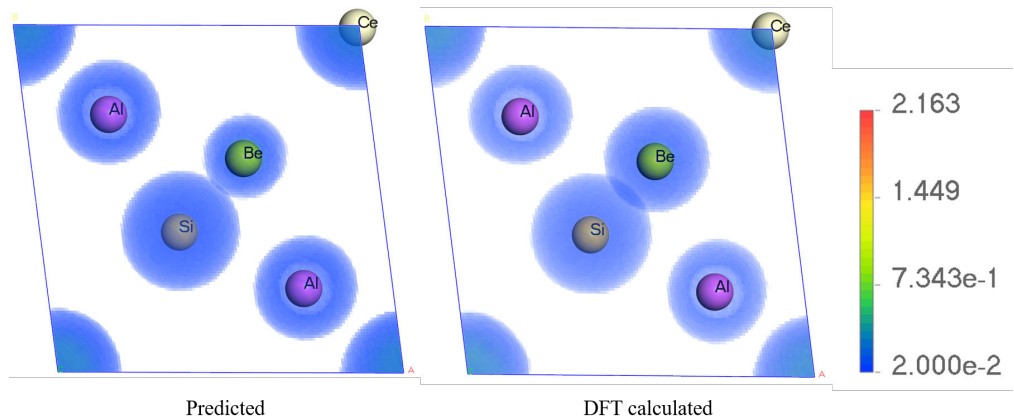

Figure 8: Case of OOD validation experiment, material code in Gnome database: ebfc3a7854.

## G.1 SAMPLE DATA

Below is a sample ECD file, demonstrating its typical layout:

```
CeBeAlSi
  1.00000000000000
    7.424433   -0.016098    2.765546
   -0.957816    7.408746    2.697069
    0.066512   -0.029102   11.747799
   Ce   Be   Al   Si
    1    1    2    1
Direct
  0.994813   0.994638   0.010597
  0.616360   0.616420   0.766230
  0.742351   0.242544   0.515314
  0.242735   0.742366   0.515387
  0.403740   0.404030   0.192473
  120  120  180
  0.23427331455E+03  0.28689571492E+03  0.44404971045E+03
  ...
  augmentation occupancies   5  33
  0.1828841E+00 -0.9408141E-01 -0.2320062E-02  0.3674660E-02 -0.1212762E-02
  0.1020137E-02 -0.4797309E-03  0.8535439E-03  0.1785949E-01  0.1289605E-02
  ...
```

## G.2 EXPLANATION

- **Crystal structure information:** The first section provides details of the crystal structure, including the system name, scaling factor, lattice vectors, element types, atomic counts, and atomic coordinates.

- **FFT grid dimensions:** The line `120 120 180` specifies the grid dimensions (`NGXF`, `NGYF`, `NGZF`) for the charge density representation.

- **Charge density data:** The charge density, scaled by the FFT-grid volume, is represented as multiple real numbers per line and continues until all `NGXF`×`NGYF`×`NGZF` values are listed.

- **Augmentation occupancies**: Additional data required for PAW calculations, if applicable.

This format ensures compatibility with visualization tools and supports detailed analysis of charge density distributions within the crystal structure.

