# OpenReview forum: "ECD: A Machine Learning Benchmark for Predicting Enhanced-Precision Electronic Charge Density in Crystalline Inorganic Materials"
_ICLR.cc/2025/Conference — ICLR 2025 Oral_

### Official Review · Reviewer_Tq9X · 2024-10-30

**Soundness:** 2
**Presentation:** 2
**Contribution:** 2
**Rating:** 6
**Confidence:** 4

**Summary:**

The author introduces a new benchmark for predicting electronic charge density for crystalline inorganic materials. The dataset includes 140,646 stable crystal geometries with medium-precision PBE functional, and high accuracy HSE from subsets of the meticulously curated geometries.

**Strengths:**

1. The computational efforts used in this study is immense, and the resulting benchmark could be a valuable asset to the ML community.
2. Interesting task on predicting the gaps across different functionals.

**Weaknesses:**

1. The work provides minimal justification for predicting electrons instead of the Kohn-Sham Hamiltonian.
2. There is no comparison with other quantum chemistry benchmarks, like QH9. This will make the paper's calculation comparable in the literature.
3. The EXP-id lacks detail on its 47 entries, which is essential for practical use. Additionally, the assessment of practical application is sparse.
4. The writing lacks cohesion, with many GPT-like passages. Figure 2 is unclear: does moving up or down the ladder imply increased simplicity?
5. The metrics are limited to MAE for charge density and acceleration ratio. Other measures, such as total energy, HOMO-LUMO gap, electronic spatial extent, or dipole moment, should be included, as this is particularly important for practical applications.

**Questions:**

1. Is there a fundamental difference between predicting the Kohn-Sham Hamiltonian and electron density? The author should discuss the computational challenges and benefits of predicting electron density, as several prior works have focused on predicting the Kohn-Sham Hamiltonian.

2. Can the MAE of electron density be compared to that of a Kohn-Sham Hamiltonian prediction model?

3. In line 528, "due to the high dimensionality of charge density data, often involving over 1,000 nodes, the complexity of handling node and edge interactions presents significant computational challenges"—what do "node" and "edge" refer to here?

4. What does the regression output look like for a given structure? A more detailed example or description of the matrix's size and shape would be helpful.

5. How does the model perform on predicting HOMO and LUMO?

6. Could the author provide a more detailed discussion on error? With an HSE error of 1 eV, what is the impact on practical applications like energy band predictions? For a usable model, is 1 eV sufficient for downstream applications?

---

> ### Author Response · Authors · 2024-11-22
> **Rebuttal by Authors - Part 1**
>
> > **W1: The work provides minimal justification for predicting electrons instead of the Kohn-Sham Hamiltonian.**
>
> We appreciate the reviewer’s observation. Our decision is justified as follows:
>
> 1.  Electronic density, a fundamental physical quantity, enables intuitive analysis of phenomena like reactivity and bonding while supporting downstream applications.
> 2. As a scalar 3D field, it is well-suited for deep neural networks, unlike the Kohn-Sham Hamiltonian, which involves non-local operators, adding prediction complexity.
> 3.  Predicting the Kohn-Sham Hamiltonian requires solving an eigenvalue problem to obtain the electronic density, where small prediction errors can be amplified, leading to unreliable results. Directly predicting the electronic density avoids this sensitivity, ensuring greater reliability and reducing computational complexity.
> 4. Leading DFT software (e.g., VASP, QE) directly outputs electronic density, supporting large-scale dataset generation for high-throughput studies.
>
> > **W2: There is no comparison with other quantum chemistry benchmarks, like QH9. This will make the paper's calculation comparable in the literature.**
>
> We thank the reviewer for the suggestion. QH9 is a dataset focused on Hamiltonian calculations for small molecular systems using atom-centered basis sets, whereas ECD targets solid-state materials and is generated using plane-wave methods to calculate charge densities. This distinction reflects the different scopes and methodologies of the two datasets.
>
> > **W3: The EXP-id lacks detail on its 47 entries, which is essential for practical use. Additionally, the assessment of practical application is sparse.**
>
> We thank the reviewer for the suggestion. We have made the 47 entries (including predictions and structure codes) available in the code repository. The limited number of entries is due to the intersection of data collected from MP, OQMD, JARVIS databases, HSE calculations, and experimental values. We plan to expand the dataset further in future work.
>
> > **W4: The writing lacks cohesion, with many GPT-like passages. Figure 2 is unclear: does moving up or down the ladder imply increased simplicity?**
>
> We thank the reviewer for the feedback. We have polished the manuscript for better cohesion. In Figure 2, the intention is to show that accuracy increases with computational complexity, and we have clarified this in the revised manuscript.
>
> > **W5: The metrics are limited to MAE for charge density and acceleration ratio. Other measures, ... as this is particularly important for practical applications.**
>
> We thank the reviewer for the suggestion. For solid-state materials, mainstream datasets like MP and OQMD primarily focus on properties such as band gaps and formation energies. Band gaps, analogous to HOMO-LUMO gaps in molecular systems, are derived from electronic structures, while formation energies are calculated from total energies. Properties like electronic spatial extent and dipole moments are mainly relevant to molecular systems and are easier to compute using tools like PySCF. However, obtaining standardized results for these properties with VASP would require future work.
>
> > **Q1: Is there a fundamental difference between predicting the Kohn-Sham Hamiltonian and electron density? The author should ...**
>
> We thank the reviewer for the suggestion. There is indeed a fundamental difference between predicting the electron density and the Kohn-Sham Hamiltonian, as highlighted in W1. We will include a detailed discussion of these differences and their computational implications in the revised manuscript.
>
> > **Q2: Can the MAE of electron density be compared to that of a Kohn-Sham Hamiltonian prediction model?**
>
> We thank the reviewer for the constructive suggestion. Both electron density and Kohn-Sham Hamiltonian prediction models serve as auxiliary tools and require integration with DFT software, such as VASP or PySCF, for further calculations. Each approach is better suited to different systems—solid materials for electron density and non-periodic small molecules for Hamiltonians. While developing Hamiltonian IO support in VASP is theoretically feasible, it is beyond the scope of this work.
>
> > **Q3: In line 528, ... what do "node" and "edge" refer to here?**
>
> We apologize for the confusion. Here, "node" refers to atoms and electrons, while "edge" represents the connections formed between them within a specified cutoff distance.

---

> > ### Author Response · Authors · 2024-11-22
> > **Rebuttal by Authors - Part 2**
> >
> > > **Q4: What does the regression output look like for a given structure? A more detailed example or description of the matrix's size and shape would be helpful.**
> >
> > The regression output for a given structure is a 3D scalar field representing the charge density, typically on a regular grid spanning the unit cell. The grid size and shape depend on the system's lattice parameters and the plane-wave cutoff used in the DFT calculation. For example, a cubic unit cell might have a grid size of $$\( N_x \times N_y \times N_z \)$$, where $$\( N_x, N_y, N_z \)$$ are determined by the resolution of the calculation. We will include a detailed example in the revised manuscript.
> >
> > > **Q5: How does the model perform on predicting HOMO and LUMO?**
> >
> > Our model is primarily designed to predict charge density rather than HOMO and LUMO energy levels, which are not explicitly targeted in this work. However, by feeding the charge density back into the DFT software, we can further calculate the electronic structure to obtain the valence band maximum (VBM) and conduction band minimum (CBM), which are analogous to the HOMO and LUMO in molecular systems.
> >
> > > **Q6: Could the author provide a more detailed discussion on error? With an HSE error of 1 eV, what is the impact on practical applications like energy band predictions? For a usable model, is 1 eV sufficient for downstream applications?**
> >
> > We thank the reviewer for this important question. An HSE error of 1 eV can have varying impacts depending on the application. For tasks like high-throughput screening, where relative trends are more critical than absolute accuracy, a 1 eV error may still be sufficient. However, for precise applications like band gap engineering or optoelectronic device design, this level of error might lead to significant deviations. In such cases, achieving an error below 0.2–0.5 eV is often required for reliable predictions. We will add a discussion on this topic in the revised manuscript.
> >
> > Thank you once again for your valuable suggestions. We have addressed your concerns and would greatly appreciate your feedback and reconsideration of the scores.

---

> > > ### Comment · Reviewer_Tq9X · 2024-11-24
> > >
> > > I thank the authors for their response. The author claims "Electron density is well-suited for deep neural networks, unlike the Kohn-Sham Hamiltonian, which involves non-local operators, adding prediction complexity." which is not empirically justified. Can author also expand on "Predicting the Kohn-Sham Hamiltonian requires solving an eigenvalue problem to obtain the electronic density, where small prediction errors can be amplified, leading to unreliable results. Directly predicting the electronic density avoids this sensitivity, ensuring greater reliability and reducing computational complexity." This claim is also not well justified without any references or experiments.

---

> > > > ### Author Response · Authors · 2024-11-24
> > > > **Further elaboration of Claim 1**
> > > >
> > > > > **Q1: The author claims "Electron density is well-suited for deep neural networks, unlike the Kohn-Sham Hamiltonian, which involves non-local operators, adding prediction complexity." which is not empirically justified**
> > > >
> > > > We thank the reviewer for their thoughtful feedback and for highlighting the need for further justification. Below, we clarify the relative complexity of predicting the electron density versus the Kohn-Sham Hamiltonian.
> > > >
> > > > The KS Hamiltonian is derived from DFT and typically computed using atomic localized basis functions (e.g., Gaussian or PySCF). These basis functions are directly defined in the local environment of each atom and exhibit the following characteristics:
> > > > - Each basis function is associated with specific atomic orbitals (e.g., 1s, 2p, 3d).
> > > > - The interaction between basis functions of different atoms is determined by specific coupling relationships, such as orbital overlap integrals and exchange-correlation effects.
> > > >
> > > > The Hamiltonian matrix elements H_{ij} ​are inherently tied to the atomic orbital interactions, leading to the following complexities:
> > > >
> > > > - Diagonal Elements: Representing local orbital energies.
> > > > - Off-Diagonal Elements: Represent the coupling strength between orbitals of different atoms, involving terms like overlap integrals and exchange integrals.
> > > >
> > > > These off-diagonal terms encode non-local interactions, requiring high-dimensional representations. Models such as QHNet[1] explicitly address this by introducing tensor-based representations for orbital couplings, and DeepH[2] leverages layers like Local Coordinate Message Passing (LCMP) to handle directional dependencies and SE(3)-equivariance. Such designs reflect the inherent complexity of modeling the Hamiltonian as a high-dimensional matrix with intricate non-local interactions.
> > > >
> > > > In contrast, electron density is a scalar field with predominantly local dependencies. Models like ChargE3Net focus on local atomic and pairwise features (e.g., distances, directions) to predict density distributions. The local nature of electron density significantly simplifies its representation compared to the Hamiltonian.
> > > >
> > > > [1]. PMLR 202:40412-40424, 2023
> > > >
> > > > [2]. Nature Computational Science volume 2, pages367–377 (2022)

---

> > > > > ### Author Response · Authors · 2024-11-24
> > > > > **Further elaboration of Claim 2**
> > > > >
> > > > > > **Q2: Can author also expand on "Predicting the Kohn-Sham Hamiltonian requires .., ensuring greater reliability and reducing computational complexity." This claim is also not well justified without any references or experiments.**
> > > > >
> > > > > Thank you for your insightful comment. We appreciate the opportunity to clarify and expand upon the statement regarding the prediction of the Kohn-Sham Hamiltonian versus directly predicting the electronic density.
> > > > >
> > > > > In Kohn-Sham Density Functional Theory (KS-DFT), the central task is to determine the ground-state electronic density $n(\mathbf{r})$ of a many-electron system. This is achieved by solving the Kohn-Sham equations, which involve constructing the Kohn-Sham Hamiltonian $\hat{H}_{KS}$ and solving a set of self-consistent single-particle equations:
> > > > >
> > > > > $$
> > > > > \hat{H}_{KS} \psi_i(\mathbf{r}) = \epsilon_i \psi_i(\mathbf{r}),
> > > > > $$
> > > > >
> > > > > where
> > > > >
> > > > > $$
> > > > > \hat{H}_{KS} = -\frac{\hbar^2}{2m_e} \nabla^2   + V_{\text{eff}}n(\mathbf{r})
> > > > > $$
> > > > >
> > > > > and $V_{\text{eff}}n(\mathbf{r})$ is the effective potential dependent on the electronic density $n(\mathbf{r})$. The electronic density is then obtained from the Kohn-Sham orbitals $\(\psi_i(\mathbf{r})\)$:
> > > > >
> > > > > $$
> > > > > n(\mathbf{r}) = \sum_{i \in \text{occ}} |\psi_i(\mathbf{r})|^2.
> > > > > $$
> > > > >
> > > > > Predicting the Kohn-Sham Hamiltonian using machine learning involves approximating
> > > > >
> > > > > $$
> > > > > V_{\text{eff}}n(\mathbf{r})
> > > > > $$
> > > > >
> > > > > directly. However, small errors in the predicted potential can be significantly amplified when solving the eigenvalue problem to obtain $\psi_i(\mathbf{r})$ and $\epsilon_i\$. This sensitivity arises because the eigenvalues and eigenfunctions are nonlinear functions of the Hamiltonian. Perturbation theory indicates that even minor inaccuracies in the Hamiltonian can lead to substantial deviations in the eigenvalues and eigenfunctions, particularly in systems with closely spaced energy levels or near-degenerate states [1, 2].
> > > > >
> > > > > This amplification of errors can result in an electronic density $n(\mathbf{r})$ that deviates considerably from the true density, leading to unreliable results. The cumulative effect of these errors makes the indirect approach of predicting the Hamiltonian less robust, as the inaccuracies are not easily mitigated through post-processing.
> > > > >
> > > > > In contrast, directly predicting the electronic density with machine learning bypasses the eigenvalue problem entirely. By formulating the problem as a direct mapping from atomic configurations to $n(\mathbf{r})$, we avoid the error-prone step of solving for the eigenvalues and eigenfunctions. This direct approach reduces sensitivity to small prediction errors, as the machine learning model can be trained to capture the complex, non-linear relationship between atomic positions and the electronic density without intermediary steps that could amplify errors.
> > > > >
> > > > > We hope this addresses your concerns and provides a clearer rationale for our approach.
> > > > >
> > > > >
> > > > > [1]. Trefethen L N, Embree M. *The behavior of nonnormal matrices and operators*. Spectra and Pseudospectra, 2005.
> > > > >
> > > > > [2]. Cancès E, Deleurence A, Lewin M. *A new approach to the modeling of local defects in crystals: the reduced Hartree-Fock case*. Communications in Mathematical Physics, 2008, 281: 129-177.

---

> ### Comment · Reviewer_Tq9X · 2024-11-25
>
> Dear Authors
>
> Thank you for your detailed response, which has partially addressed my concerns. I am also curious about the details of EquiChargE3Net. For a given structure, since your output is NxNyNz, how do you construct this from the invariant and equivariant node/edge features?

---

> > ### Author Response · Authors · 2024-11-26
> > **Details of EquiChargE3Net Model**
> >
> > > **Q1: xx. For a given structure, since your output is NxNyNz, how do you construct this from the invariant and equivariant node/edge features?**
> >
> > We sincerely appreciate the reviewers' questions and apologize for not clearly explaining the equivariant features. In the ChargeE3Net model, the input data is as follows:
> >
> > - Each atom's atomic number: $ \{z_1, ..., z_N\} \in \mathbb{N} $
> > - Each atom's three-dimensional position: $ \{\vec{r}_1, ..., \vec{r}_N\} \in \mathbb{R}^3 $
> > - Probe position: $ \{\vec{r}_1, ..., \vec{r}_M\} \in \mathbb{R}^3 $
> > - Unit Cell: $ C \in \mathbb{R}^{3 \times 3} $
> >
> > These data points provide complete graph structure information. The number of Probe is NGX×NGY×NGZ. To reduce computational cost, it is also possible to perform sampling from NGX×NGY×NGZ. Each atom serves as a node in the atomic graph, while the grid points define another graph structure where each point also serves as a node. The edges are constructed based on proximity (e.g., atoms or grid points within a specific cutoff radius form edges). Therefore, we define represented nodes as:
> > - Atomic nodes:  Atomic features $ a_i \in \mathbb{R}^F $ (obtained through learning).
> >
> > - Probe nodes: Grid invariant features, representing local atomic property information.
> >
> > Then we construct graph edge features as:
> > - Distance features: Features derived from the distance between nodes, which are input into the filtering function.
> >
> > - Directional features: Features derived from the directional vector between nodes, used for rotational or spatial equivariant calculations.
> >
> > When constructing edge features, we consider scenarios where an atom’s distance to a probe grid point is smaller than a given threshold (e.g., 4 Å), at which point an edge is formed. Similarly, edges between atom-atom or probe-probe nodes can also be constructed, depending on specific needs.
> >
> > If the input to the model includes only edge distance information, it corresponds to the InvChargE3Net model, where the message passing method is based on the SchNet framework[1]. If the model considers both distance and edge vector information, it corresponds to the EquiChargE3Net model, where the message passing method is based on the PaiNN framework[2].
> >
> > The final model output is the features of the probes, representing the electron density, with dimensions of NGX×NGY×NGZ. Of course, if we sample probes from NGX×NGY×NGZ, post-processing method like interpolation is required to expand the probes back to NGX×NGY×NGZ.
> >
> > We hope to address your concerns and look forward to receiving your feedback.
> >
> > [1]. The Journal of Chemical Physics, 2018, 148(24).
> >
> > [2]. International Conference on Machine Learning. PMLR, 2021: 9377-9388.

---

> > > ### Comment · Reviewer_Tq9X · 2024-11-26
> > >
> > > If I understand correctly, the method is pretty similar to that of [1], right?
> > >
> > > [1]: Wang et al. Neural P3M: A Long-Range Interaction Modeling Enhancer for Geometric GNNs. NeurIPS 2024.

---

> > > > ### Author Response · Authors · 2024-11-26
> > > >
> > > > Your understanding is correct. This paper adopts the approach of incorporating mesh points alongside atoms to predict molecular system properties such as energy and forces. We predict the electronic charge density by modeling the interactions between the probe points on the mesh and the atoms.

---

> > > > > ### Comment · Reviewer_Tq9X · 2024-11-26
> > > > >
> > > > > Can you elaborate on the differences between the probe points and mesh points? So probe points are just downsampled mesh points, right?

---

> > > > > > ### Author Response · Authors · 2024-11-26
> > > > > >
> > > > > > We have verified that the mesh points used in Equation 10 [1] are the same as the probe points defined in Charge3Net.
> > > > > >
> > > > > > [1]: Wang et al. Neural P3M: A Long-Range Interaction Modeling Enhancer for Geometric GNNs. NeurIPS 2024.

---

> > > > > > > ### Comment · Reviewer_Tq9X · 2024-11-26
> > > > > > >
> > > > > > > Thank you for your response. I am increasing my score to 6.

---

### Official Review · Reviewer_NK1Z · 2024-11-02

**Soundness:** 4
**Presentation:** 3
**Contribution:** 4
**Rating:** 8
**Confidence:** 5

**Summary:**

This paper introduces the Electronic Charge Density (ECD) dataset, a benchmark designed for predicting electronic charge densities in crystalline inorganic materials. The dataset comprises 140,646 stable crystal structures computed with PBE functional, alongside a subset of 7,147 structures calculated with HSE functional. Using the ChargE3Net model as a baseline, the authors evaluate its performance across multiple tasks, including prediction accuracy on the ECD dataset and the acceleration of DFT computations. Overall, this paper makes a valuable contribution to machine learning in quantum chemistry and computational materials science, and I recommend acceptance conditional on addressing the concerns outlined below.

**Strengths:**

* Originality: Good. The data generated in this work is indeed original.
* Quality: Good. The functionals PBE and HSE used in this work are indeed the widely accepted ones to compute crystalline inorganic materials. Thus, the data generated in this work are of good qualities.
* Clarity: The paper is well-structured, with a clear explanation of the dataset’s construction, tasks, and evaluation metrics.
* Significance: This is a significant contribution to this area. Due to the computational cost to carefully converge the charge densities, it is not easy to generate or to find publicly available data sets. Thus, this work will clearly benefit the researchers in this area.

**Weaknesses:**

* Limited High-Precision Data: I totally understand that the HSE calculations are much more expensive, but it would be much better if the authors can keep increasing the number of calculations done with HSE functional.

**Questions:**

Although I generally believe this paper is a good contribution to this area, I still have the following concerns and I hope the authors may address:
1. **Codes and Data Availability**: The links to the workflows and datasets are no longer valid, thus I could not review them. It would be nice if the authors provide new links to them.
2. **Computational Setup**:

    a) Why did the authors decide to mix charge densities with PBE and PBE+U functionals? Are you sure that the charge densities at these two levels are compatible to be mixed up? More information for you to consider: In public datasets like MP, they mixed PBE and PBE+U functionals mostly because they justified that they can align the formation energies of materials with further compatibility corrections. However, here you are computing charge densities, so I am skeptical if these two levels of theories should be mixed up.

    b) The choice of hybrid functional is not justified. There are multiple hybrid functionals available on the market, including PBE0, B3LYP, the HSE you are using, etc. Why did you pick up HSE as your functional? In addition, you should specify which version of HSE you used, HSE03 or HSE06?

    c) Did you apply any convergence test on the charge densities with respect to the energy cutoff and Brillouin zone sampling? My impression is that the 520eV cutoff you are using might not have converged the charge densities.

3. **Data Storage Schema**: In large-scale calculations used for machine learning, it would be nice if you include the information of the data storage schema in your supplementary information. For example,  what information you stored and what units for each of the information.

4. **Experimental Setups**

    a) **Question on OOD evaluation using GNoME dataset**: The authors *somehow* selected 2k materials from GNoME dataset, which does not make sense to me that these are naturally *out of distribution*. I suggest the authors conduct structure matching test to see how many in the 2k materials are truly out of distribution. Otherwise, this is not good enough to claim that this is an evalutation on OOD materials.

    b) There is a property that the charge densities have to satify:
    $$\int \rho(\mathbf{r}) \mathrm{d}\mathbf{r} = \text{No. of electrons}$$
    How did you enforce this in your model?

    c) The evaluation of the prediction of charge densities is based on the MAE with respect to the ground truth, which makes sense in terms of developing ML models. However, I am interested to see how much error in total energy or band gap if the predicted charge densities are put back to DFT iterations. These are the realistic application of the charge densities, otherwise one can not really infer any useful information from predicted charge densities. The authors may consider commenting on this or conducting one or a few examples to show the performance.

---

> ### Author Response · Authors · 2024-11-22
> **Rebuttal by Authors - Part 1**
>
> > **W1: Limited High-Precision Data: I totally understand that the HSE calculations are much more expensive, but it would be much better if the authors can keep increasing the number of calculations done with HSE functional.**
>
> Thank you for the constructive feedback. We agree on the importance of expanding the dataset with high-precision HSE calculations. Moving forward, we will leverage supercomputing resources and adopt an active learning approach with uncertainty quantification to strategically select rare and critical structures, maximizing the value of high-precision data while optimizing computational resources.
>
> > **Q1: Codes and Data Availability: The links to the workflows and datasets are no longer valid, thus I could not review them. It would be nice if the authors provide new links to them.**
>
> Thank you for carefully checking the links to the workflows and datasets. We have reviewed and updated the links accordingly to ensure accessibility.
> DFT workflow: https://anonymous.4open.science/r/DFTflow-635A,
> Bechmark: https://anonymous.4open.science/r/ECDBench-037F.
> Please feel free to reach out if you encounter any further issues.
>
> > **Q2: Computational Setup: a) Why did the authors decide to mix charge densities with PBE and PBE+U functionals?**
>
> Thank you for your suggestion. In our dataset, the +U correction is systematically applied based on element types, and its effect can be learned by AI models during training. While the PBE functional alone has an error exceeding 30%, the empirical +U correction improves accuracy to some extent. For large-scale databases, the +U correction offers a practical balance between computational cost and accuracy, making it suitable for generating properties like bandgap and formation energy useful for materials discovery.
>
> > **Q2: Computational Setup: b) The choice of hybrid functional is not justified. Why did you pick up HSE as your functional? In addition, you should specify which version of HSE you used, HSE03 or HSE06?**
>
> Thank you for your insightful comment. Among the available hybrid functionals, HSE is widely adopted in materials science due to its ability to reasonably capture electronic structure properties, especially bandgap values[1,2], compared to PBE and other functionals. Regarding the specific version, we used HSE06, as described in line 260 of the manuscript. In the revised version, we will make this explicitly clear as HSE06.
>
> [1]. Computational Materials Science, 2024, 239: 112956.
> [2]. International Conference on Simulation of Semiconductor Processes and Devices (SISPAD). IEEE, 2021: 133-137.
>
> > **Q2:Computational Setup: c) Did you apply any convergence test on the charge densities with respect to the energy cutoff and Brillouin zone sampling? My impression is that ...**
>
> Thank you for your suggestion. The energy cutoff is typically set to at least 1.2 times the ENMAX value from the POTCAR file. In our case, all elements meet this criterion except He (ENMAX=478.9) and Li (ENMAX=499). Raising the cutoff above 1.2 times ENMAX for these elements would significantly increase computational cost, so we chose a cost-effective value of 520 eV. This approach is widely used in other DFT databases like MP and JARVIS. For Brillouin zone sampling, we started with a coarse grid and iteratively refined it up to five steps, as detailed in our DFT workflow. The final dense grid ensures convergence and balances accuracy with efficiency.
>
> > **Q3: In large-scale calculations used for machine learning, it would be nice if you include the information of the data storage schema in your supplementary information. For example, what information you stored and what units for each of the information.**
>
> Thank you for your suggestion. The raw charge density data is around 10TB, reduced to 3.3TB after compression. Our training code supports direct reading from the compressed files for efficient handling. In the revised manuscript, we have added details on the data storage schema, including the stored information and its structure for seamless machine learning integration.

---

> > ### Author Response · Authors · 2024-11-22
> > **Rebuttal by Authors - Part 2**
> >
> > > **Q4: Experimental Setups:a) Question on OOD evaluation using GNoME dataset: The authors somehow selected 2k materials from GNoME dataset, which does not make sense to me that these are naturally out of distribution. I suggest ...**
> >
> > Thank you for your constructive feedback. We have conducted similarity calculations for the 2k materials from the GNoME dataset. The results indicate that there are no similar structures between these materials and the 140k structures in our dataset. Given the large computational cost, we perform the following steps: first, check if the chemical formula and atom count of the structure in GNoME are present in the ECD. If they are, we then proceed with similarity calculations using pymatgen. This confirms that the selected materials are truly out of distribution. We will include the detailed methodology and results of the similarity calculations in the revised manuscript to further substantiate our claim.
> >
> > > **Q4: Experimental Setups:b)There is a property that the charge densities have to satify ..,How did you enforce this in your model?**
> >
> > Thank you for your suggestion. The model's training labels come from CHGCAR files, which include electron density values on a defined grid $$\(N_x, N_y, N_z\)$$. During testing, the grid must also be provided to calculate the MAE between predicted and actual values. While the theoretical constraint $$\(\int \rho(\mathbf{r}) d\mathbf{r} = \text{Number of electrons}\) $$is essential, it was not explicitly enforced in the model during training or inference. Our dataset may provide opportunities to explore and enforce such physical constraints in future developments.
> >
> > > **Q5: Experimental Setups:c)The evaluation of ... However, I am interested to see how much error in total energy or band gap if the predicted charge densities are put back to DFT iterations. ... The authors may consider commenting on this or conducting one or a few examples to show the performance.**
> >
> > Thank you for your constructive comments. The table below addresses your concerns: In this experiment, we fed the predicted charge densities back into DFT to predict bandgap properties and compared the results with end-to-end ML methods, PBE, and HSE calculations, using experimental values as the benchmark. The results demonstrate that the MAE performance of the directly predicted charge densities approaches that of HSE functional calculations. While it is slower than the end-to-end property prediction models in terms of speed, it offers improved accuracy and greater generalizability.
> >
> > | Model/Dataset     | MAE (eV)          | Time per structure |
> > |--------------------|-------------------|---------------------|
> > | **ML Models**      |                   | (s)                 |
> > | CGCNN             | 1.45 Chen et al. (2019) | 1.5              |
> > | MEGNet            | 1.36 Chen et al. (2019) | 1.34             |
> > | CrystalNet        | 1.19 Chen et al. (2022) | 1.67             |
> > | CrystalNet-TL     | 0.70 Chen et al. (2022) | 1.58             |
> > | **ECD-Based Models** |                   | (min)               |
> > | ECD-PBE           | 1.17              | 14.4               |
> > | ECD-PBE_HSE       | 0.65              | 14.4               |
> > | **PBE-Based Models** |                   | (min)               |
> > |MP                 | 1.38 Choudhary et al. (2018)  | -          |
> > |Matgen             | 1.21 Chen et al. (2022)       |24.5        |
> > |AFLOW              | 1.20 Choudhary et al. (2018)  | -          |
> > |OQMD               |1.09 Choudhary et al. (2018)   | -          |
> > | **HSE-Based Models** |                   | (min)               |
> > | HSE               |0.41 Choudhary et al. (2018)   |228.1       |
> >
> > Thank you again for your valuable suggestions. We believe we have addressed your concerns and kindly hope for your feedback and reconsideration of the scores.

---

> > > ### Comment · Reviewer_NK1Z · 2024-11-24
> > > **Further explanation requested**
> > >
> > > Dear authors,
> > >
> > > Thank you for addressing my previous concerns. However, several important points require further clarification:
> > > * **Data Schema**: I need to clarify my earlier question regarding the data schema. Specifically, I am interested in understanding how data entries are structured and organized within your database. Given that this paper introduces a significant dataset, documenting the data organization would be valuable for potential users to understand and effectively utilize the information contained within.
> > > * **Electron Density Integration**: The conservation of electron density remains a concern. Since this constraint was not enforced in your implementation, please address how this might affect the model's accuracy and reliability. If possible, additional experiments demonstrating the impact of this decision would strengthen your manuscript significantly.
> > > * **Experimental Validation Approach for Q5**: Your current benchmarking methodology, which compares results against experimental values, requires reconsideration. The PBE functional is well-known to underestimate band gaps in materials. Since your study aims to evaluate whether predicted electron density can accelerate calculations or improve electronic structure predictions, it would be more appropriate to calculate Mean Absolute Errors (MAEs) against PBE or HSE results, depending on your specific objectives.
> > >
> > > Given these outstanding concerns, my recommendation regarding this manuscript remains unchanged.

---

> ### Author Response · Authors · 2024-11-25
> **Response to Data Schema**
>
> > **Q1：Data Schema: ...how data entries are structured and organized within your database. ..**
>
> Thank you for your valuable suggestions. We appreciate your insightful comments, which have significantly contributed to improving the quality of our paper. Regarding the question on the data schema, we provide the following details:
>
> The charge density data in our dataset is stored in a text format, with each file generally less than 1 GB in size. ECD files are named according to their material IDs, and detailed information is provided as follows:
> ```
> System name
> Scaling factor
> Lattice vector 1
> Lattice vector 2
> Lattice vector 3
> Element names and atom counts
> Coordinate system (Direct)
> Atomic coordinates
> FFT grid dimensions
> Charge density data (flattened)
> Augmentation occupancies (if applicable)
> ```
> The core information in the file consists of three sections, namely:
>
> - Crystal structure information: The initial section provides essential information about the crystal structure, including the system name, scaling factor, lattice vectors, element types, atomic counts, and atomic coordinates.
> - FFT grid dimensions: A specific line defines the grid dimensions, e.g., 120 120 180, corresponding to the charge density representation's FFT-grid size (NGXF, NGYF, NGZF).
> - Charge density data: The charge density values, scaled by the FFT-grid volume, are represented as a sequence of real numbers, listed continuously until all NGXF × NGYF × NGZF values are included.
> - Augmentation occupancies: If applicable, additional data required for PAW (Projector Augmented Wave) calculations is included at the end.
>
> This text-based format ensures compatibility with common visualization tools and facilitates detailed analysis of charge density distributions in the crystal structure. The organization enables easy integration with computational pipelines and supports advanced analyses, such as structure-property relationships or electronic structure evaluations.
>
> We hope this explanation clarifies the structure and organization of the dataset.

---

> > ### Author Response · Authors · 2024-11-25
> > **Response to Electron Density Integration**
> >
> > > **Q2: Electron Density Integration: The conservation of electron density remains a concern. ...**
> >
> > Thank you very much for the reviewer’s suggestion. The conservation of electron density is indeed a critical concern, and we appreciate your valuable feedback. Below, we provide additional evidence to demonstrate that although our model does not explicitly enforce this constraint at the model level, the inferred grid density can mitigate potential impacts on downstream property predictions.
> >
> > In our training and inference pipeline:
> > - The grid point numbers NGX, NGY, and NGZ are directly obtained from the `CHGCAR` file, as written by VASP. These grid numbers dictate the spatial resolution of the grid and are used by the model to determine the number of grid points and, subsequently, the electron density at each point.
> > - It is worth noting that when the resolution is too low, discrepancies may arise between the calculated total electron count (via grid integration) and the theoretical valence electron count of the constituent atoms.
> >
> > For datasets without explicit VASP annotations of NGX, NGY, and NGZ, we leverage the computational formula provided by VASP to estimate these grid parameters. The detailed formula is as follows:
> >
> > $$
> > NGX = \left\lceil \frac{a^* \cdot \sqrt{\text{ENCUT}}}{\pi} \right\rceil, \quad
> > NGY = \left\lceil \frac{b^* \cdot \sqrt{\text{ENCUT}}}{\pi} \right\rceil, \quad
> > NGZ = \left\lceil \frac{c^* \cdot \sqrt{\text{ENCUT}}}{\pi} \right\rceil
> > $$
> >
> > Where:
> > - $a^*, b^*, c^*$ are the reciprocal lattice vectors' lengths (unit: $\text{\AA}^{-1}$).
> > - $\text{ENCUT}$ is the energy cutoff value (unit: $\text{eV}$).
> > - $\lceil  \rceil$ represents the ceiling function to ensure integer grid points.
> >
> > To mitigate any residual discrepancies caused by grid resolution. We ensure that the inferred grid density closely approximates the theoretical resolution, as defined by the VASP fine grid size, which is typically 8 times the theoretical resolution. This reduces errors in total electron count and minimizes the influence on downstream property predictions.
> >
> > The experiments we conducted are as follows: Using AgO material (ICSD code: 670509), based on its elemental composition and lattice parameters, we calculated NGX=NGY=NGZ=7. We then analyzed the relationship between different grid densities and total energy:
> > ```
> > 7×7×7 resulted in -12.5992 eV;
> > 14×14×14 resulted in -5.0207 eV;
> > 28×28×28 resulted in -7.2872 eV;
> > 56×56×56 resulted in -7.2872 eV;
> > 84×84×84 also resulted in -7.2872 eV.
> > ```
> > It can be observed that setting an 8-fold grid density ensures stable energy calculations.
> >
> > By using the inferred grid parameters from either VASP outputs or the computational formula, our model ensures sufficient resolution to capture the conservation of electron density effectively. We believe this approach strengthens the reliability of our method, even in the absence of explicit conservation constraints.

---

> > > ### Author Response · Authors · 2024-11-25
> > > **Response to Experimental Validation Approach for Q5**
> > >
> > > > **Q3：Experimental Validation Approach for Q5:**
> > >
> > > We sincerely appreciate the reviewer’s insightful comments regarding our benchmarking methodology. Following the suggestion, we have updated the tables in our manuscript to calculate the Mean Absolute Errors (MAEs) against HSE results instead of experimental values. This adjustment aligns with the well-known limitations of the PBE functional in underestimating band gaps and ensures a more appropriate comparison given our study's objectives of evaluating predicted electron densities to improve electronic structure predictions and accelerate calculations.
> > >
> > > The updated results are presented in the revised manuscript as follows:
> > > | Model/Dataset     | MAE (eV)          | Time per structure |
> > > |--------------------|-------------------|---------------------|
> > > | **ML Models**      |                   | (s)                 |
> > > | CGCNN             | 1.13 Chen et al. (2019) | 1.5              |
> > > | MEGNet            | 1.06 Chen et al. (2019) | 1.34             |
> > > | CrystalNet        | 0.89 Chen et al. (2022) | 1.67             |
> > > | CrystalNet-TL     | 0.42 Chen et al. (2022) | 1.58             |
> > > | **ECD-Based Models** |                   | (min)               |
> > > | ECD-PBE           | 0.88              | 14.4               |
> > > | ECD-PBE_HSE       | 0.39              | 14.4               |
> > > | **PBE-Based Models** |                   | (min)               |
> > > |MP                 | 1.05 Choudhary et al. (2018)  | -          |
> > > |Matgen             | 0.91 Chen et al. (2022)       |24.5        |
> > > |AFLOW              | 0.90 Choudhary et al. (2018)  | -          |
> > > |OQMD               |0.80 Choudhary et al. (2018)   | -          |
> > >
> > > We greatly appreciate the reviewer’s constructive suggestions and hope our response adequately addresses your concerns.

---

> > > > ### Comment · Reviewer_NK1Z · 2024-11-27
> > > >
> > > > Thank you for your effort in providing the explanation. I raised my recommendation to 8.

---

### Official Review · Reviewer_4VpH · 2024-11-03

**Soundness:** 3
**Presentation:** 3
**Contribution:** 3
**Rating:** 6
**Confidence:** 4

**Summary:**

The paper introduces a benchmark dataset ECD to enhance machine learning models for predicting electronic charge densities for crystalline materials. ECD includes over 140K crystal structures with medium-precision PBE data and 7147 high-precision HSE data entries. The authors propose tasks for training, fine-tuning, and out-of-distribution testing to improve predictive accuracy and speed up DFT calculations, using ChargE3Net to demonstrate significant improvement. This dataset offers a robust foundation for advancing scalable, high-accuracy electronic structure predictions in materials science.

**Strengths:**

1. The ECD dataset is a substantial and valuable contribution, offering over 140K PBE-calculated crystal structures and 7147 HSE-calculated structures, providing a robust foundation for electronic charge density prediction.

1. The paper conducts a comprehensive benchmarking of multiple models, assessing their performance across various tasks and metrics, including both in-distribution and out-of-distribution scenarios, which enhances the reliability and applicability of the findings.

1. The paper demonstrates that fine-tuning models trained on PBE data with high-precision HSE data improves predictive accuracy. This approach can guide future research on leveraging high-cost data sparingly to optimize model performance.

1. The paper highlights the model’s performance on wet-lab experimental data and out-of-distribution data, validating its potential for applications in real-world materials discovery and screening.

**Weaknesses:**

1. The description of ECD-HSE-id seems missing in Section 2.5. I assume that the data split is consistent with ECD-PBE HSE-id and ECD-PBE HSE tuning-id, but it would be great if the authors could clarify that.

1. ChargeE3Net surpasses its invariant counterparts and other non-equivariant models marginally but significantly increases the computation complexity, thus increasing the time of cost for computation. For example, comparing CrystalNet-TL to ECD-PBE HSE, the MAE is 0.70 vs 0.65 while the time per structure is 1.58s vs 14.4min.

1. In the OOD performance section, while the authors demonstrate the model's generalization, there is minimal analysis of failure cases or instances where the model struggled. Understanding these limitations in detail would be valuable.

1. While the ECD dataset is a major contribution, data augmentation techniques or other methods to improve model generalization are not discussed. Addressing potential improvements for model robustness could make the dataset more valuable for future research.

**Questions:**

1. ChargeE3Net is compared to other models in Table 2, whereas the detailed description for invDeepDFT, DeepDFT, and invChargeE3Net is missing. Can the authors justify the choice of using DeepDFT for comparison and elaborate on why it is an appropriate baseline? Besides, I guess invChargeE3Net is the invariant version of ChargeE3Net, but it would be better to clarify that in the manuscript.

1. Given the experimental results, it seems that equivariant models do not significantly decrease the errors in ECD prediction. Considering the lower computation complexity of invariant models, equivariant models may not have an edge over invariant models. Can the authors provide the DFT calculation acceleration of invChargeE3Net and make a comprehensive comparison between invariant and equivariant models taking account of both accuracy and speed?

1. Given the results in Table 3, the acceleration from ML-predicted densities is moderate as the achieved ratio is around 0.7. Furthermore, the achieved ratio is around one order of magnitude larger than the optimal ratio. Can the authors provide insight of how to further improve the acceleration or discuss what factors prevent the model from achieving the lower bound on the number of steps?

---

> ### Author Response · Authors · 2024-11-22
>
> > **W1：The description of ECD-HSE-id seems missing in Section 2.5. I assume that the data split is consistent with ECD-PBE HSE-id and ECD-PBE HSE tuning-id, but it would be great if the authors could clarify that.**
>
> We sincerely thank the reviewer for pointing out this issue. We have now provided additional clarification and detailed explanations regarding ECD-HSE-id in the manuscript.
>
> > **W2: ChargeE3Net surpasses ... For example, comparing CrystalNet-TL to ECD-PBE HSE, the MAE is 0.70 vs 0.65 while the time per structure is 1.58s vs 14.4min.**
>
> We fully agree with the reviewer that ChargeE3Net increases computational cost compared to end-to-end models like CrystalNet-TL. However, as an auxiliary model designed to accelerate DFT rather than replace it, ChargeE3Net prioritizes generalization and stability, ensuring consistent support across diverse materials. Despite its higher cost, its effectiveness in improving SCF convergence justifies its role as a valuable complement to DFT methods.
>
> > **W3: In the OOD performance section,... there is minimal analysis of failure cases or instances where the model struggled. Understanding these limitations in detail would be valuable.**
>
> We sincerely thank the reviewer for their constructive suggestions. In response, we have included a failure case in the appendix for further clarification and analysis (code: ebfc3a7854). We present a comparison plot of the predicted results and DFT results, where the visualization reveals that Charge3Net predicts the electron count of certain atoms, such as Be and Ce, to be inconsistent with the target atom.
>
> >  **W4: While the ECD dataset is a major contribution, data augmentation techniques or other methods to improve model generalization are not discussed.**
>
> We sincerely thank the reviewer for highlighting this important point. we recognize the value of data augmentation and other techniques in improving model robustness. Specifically:
> a). Incorporating methods such as Bayesian neural networks or ensemble models could enhance robustness by quantifying prediction uncertainties, particularly for out-of-distribution data.
> b). Active learning strategies could be employed to expand the HSE dataset in a more rational and efficient manner. By identifying data points with low uncertainty (e.g., using predictive uncertainty metrics or disagreement among ensemble models), additional HSE calculations can be targeted for these specific cases.
>
> > **Q1: ChargeE3Net is compared ... the detailed description for invDeepDFT, DeepDFT, and invChargeE3Net is missing. Can the authors justify the choice of using DeepDFT for comparison and elaborate on why it is an appropriate baseline?**
>
> We sincerely thank the reviewer for pointing out these important aspects. We provided detailed explanations for these models in the appendix. We selected DeepDFT as a baseline because it is one of the most representative equivariant models for charge density prediction, demonstrating competitive performance in this domain. It serves as an appropriate benchmark for evaluating the efficacy of ChargeE3Net due to their similar architectural design and shared goal of improving SCF convergence in DFT calculations.
>
> > **Q2: Given the experimental results ... Can the authors provide the DFT calculation acceleration of invChargeE3Net and make a comprehensive comparison between invariant and equivariant models taking account of both accuracy and speed?**
>
> We appreciate the reviewer’s thoughtful comments. We have supplemented the manuscript with the DFT calculation acceleration results for invChargeE3Net, as shown in the table below.
>
> | Model/Dataset     | MAE (eV)          | Time per structure |
> |--------------------|-------------------|---------------------|
> | ECD-PBE           | 1.17              | 14.4               |
> | ECD-PBE_HSE       | 0.65              | 14.4               |
> | ECD-PBE (inv)     | 1.28                | 12.1                 |
> | ECD-PBE_HSE (inv) | 0.73                | 12.1                 |
> ...
> We can observe that the performance of invChargeE3Net in terms of MAE has deteriorated, while the speed has not noticeably improved.
>
> > **Q3: Given the results in Table 3 ... Can the authors provide insight of how to further improve the acceleration or discuss what factors prevent the model from achieving the lower bound on the number of steps?**
>
> We thank the reviewer for this insightful observation. The following physical constraints related to electronic charge density may help improve the accuracy of the model:
>
> a).  The charge density must remain non-negative throughout the system:
> $$
>    \rho(r) \geq 0
> $$
> b). The integral of the charge density over the entire system must equal the total number of electrons, ensuring charge conservation:
>
>    $$
>    \int \rho(r) \, d^3r = N_{\text{electrons}}
>    $$
>
> These constraints help maintain the physical validity, accuracy, and consistency of predictive models for charge density.

---

> > ### Comment · Reviewer_4VpH · 2024-11-24
> >
> > Thank the authors for the response. The rebuttal has addressed most of my concerns. Regarding W2, the authors claim that "... as an auxiliary model designed to accelerate DFT rather than replace it, ChargeE3Net prioritizes generalization and stability, ensuring consistent support across diverse materials". For sure, I'm not expecting ChargeE3Net to replace DFT. My concern is that ChargeE3Net does not surpass the best baseline model significantly while requiring much more computation time. It would be impressive if the model could actually be generalized to diverse materials, whereas I didn't find clear evidence from the manuscript.

---

> > > ### Author Response · Authors · 2024-11-27
> > >
> > > > **Q1: My concern is that ChargeE3Net does not surpass the best baseline model significantly while requiring much more computation time. It would be impressive if the model could actually be generalized to diverse materials, ...**
> > >
> > > We sincerely appreciate the reviewer’s suggestions. We acknowledge the concern raised regarding the achieve ratio of ChargE3Net, which is approximately 70% in Table 3, and the significant discrepancy compared to the optimal ratio. This difference may be attributed to the inherent distinctions in convergence accuracy between AI models and DFT methods on both GPU and CPU platforms. However, the dataset used in our model primarily comes from the Matgen platform, which includes ICSD data. The distribution of element types and atomic numbers closely mirrors that of real materials (as shown in Fig. 3). As a result, the trained model is capable of being applied across a broad range of materials, leading to a reduction in DFT iteration steps.
> > >
> > > We highlight two valuable potential applications for discussion:
> > >
> > > - Large-scale DFT Database Construction: As exemplified by the Materials Project and the recent OMat24 from META, the Materials Project [1] contains over 100,000 DFT calculations, consuming approximately 100 million CPU hours, while OMat24 has performed 110 million DFT calculations, utilizing over 400 million CPU hours. By using our model to initialize charge density for DFT calculations, we anticipate a potential reduction of up to 30% in computational cost, which is a significant saving. Importantly, this reduction would require no modifications to the DFT code or input file parameters.
> > >
> > > - Extension to Larger Systems: For simpler systems, such as those with a single unit cell, once satisfactory accuracy is achieved, our model can be extended to larger supercell structures due to the consistent atomic environments they share. This approach is supported by experiments from DeepH [3], and is also applicable to charge density predictions. Particularly, DFT calculations are typically limited to small atomic systems (fewer than 20 atoms) for HSE06 band structures, but our model can expand the scope to larger systems.
> > >
> > > We intend to explore further valuable applications in the future.
> > >
> > > Additionally, regarding your concern about computation time, VASP has excellent scalability. The computation time shown in Table 5 was obtained on a single computing node (with 24 CPU cores in total). Using more CPUs [4] or utilizing GPU acceleration[5] can further reduce the computation time.
> > >
> > > [1]. Jain A, Montoya J, Dwaraknath S, et al. The materials project: Accelerating materials design through theory-driven data and tools[J]. Handbook of Materials Modeling: Methods: Theory and Modeling, 2020: 1751-1784.
> > >
> > > [2]. Barroso-Luque L, Shuaibi M, Fu X, et al. Open Materials 2024 (OMat24) Inorganic Materials Dataset and Models[J]. arXiv preprint arXiv:2410.12771, 2024.
> > >
> > > [3]. Li H, Wang Z, Zou N, et al. Deep-learning density functional theory Hamiltonian for efficient ab initio electronic-structure calculation[J]. Nature Computational Science, 2022, 2(6): 367-377.
> > >
> > > [4]. Kondratyuk N, Smirnov G, Agarkov A, et al. Performance and scalability of materials science and machine learning codes on the state-of-art hybrid supercomputer architecture[M]//Russian Supercomputing Days. Cham: Springer International Publishing, 2019: 597-609.
> > >
> > > [5]. Hacene M, Anciaux‐Sedrakian A, Rozanska X, et al. Accelerating VASP electronic structure calculations using graphic processing units[J]. Journal of computational chemistry, 2012, 33(32): 2581-2589.

---

> > > > ### Comment · Reviewer_4VpH · 2024-11-27
> > > >
> > > > Thanks for the response. I probably didn't clarify my point clearly. The 'baseline' I was talking about is the ML baseline models in Table 5, e.g. CrystalNet-TL which shows MAE result close to ECD-PBE HSE (0.70 vs 0.65) but the required time is much lower than ECD-PBE HSE.

---

### Official Review · Reviewer_dhoo · 2024-11-04

**Soundness:** 3
**Presentation:** 3
**Contribution:** 2
**Rating:** 6
**Confidence:** 3

**Summary:**

Paper proposes an electronic charge density dataset of 140,646 crystals using DFT PBE functional and 7,147 crystals with HSE functional. Paper trains the ChargE3Net on PBE dataset and with and without fine-tuning on HSE dataset, and compares with original ChargE3Net on MP data. Paper also shows that using the charge density predicted by ChargE3Net (as initialization) can accelerate DFT calculations. Paper compares band gaps predicted by their model with other published results.

**Strengths:**

Paper is clearly written. Paper presents dataset which was generated by significant computation expense. The problem of materials design and prediction is significant.

**Weaknesses:**

Charge density data is high memory, and may be computationally expensive to train ML models. (authors admit this in limitations section). The optimal vs achieved ratio of Table 3 seem quite far from each other- it seems to indicate that initializing dft calc with ML charge density model prediction does not improve efficiency as much as one would like.

**Questions:**

what is invChargE3Net in Table 2? could authors provide a clear definition or explanation of invChargE3Net in the paper, as this term is not defined.

What is ECD-HSE-id in Table 2? could authors clearly define or explain ECD-HSE-id in the paper, as this term is not explained.

Reviewer would suggest to authors to include a detailed description of structure of each dataset entry, possibly in appendix, if not already included.

Could authors provide more context about the Choudhary 2018 HSE-based dataset model?, including reasons for its performance characteristics, such as its error and time performance.

For Table 5, are there error bars? I am assuming the eV is band gap error, what is the acceptable error for band gap prediction?

What is reason for discrepancy between optimal and achieved ratio in Table 3, and how was optimal ratio determined?

---

> ### Author Response · Authors · 2024-11-22
>
> > **W1：Charge density data is high memory, and may be computationally expensive to train ML models. ... it seems to indicate that initializing dft calc with ML charge density model prediction does not improve efficiency as much as one would like.**
>
> We thank the reviewer for the feedback. While charge density data is memory-intensive, it is crucial for accelerating DFT calculations. The gap between the optimal and achieved ratios in Table 3 highlights the limitations of prediction accuracy. Nonetheless, ML-predicted charge densities significantly reduce SCF iterations compared to traditional methods. Future work will focus on improving model precision to narrow this gap.
>
> > **Q1: What is invChargE3Net in Table 2? could authors provide a clear definition or explanation of invChargE3Net in the paper, as this term is not defined.**
>
> Thank you for your valuable suggestion. We followed the methodology of DeepDFT[1] to implement invChargE3Net and EquiChargE3Net to enable a fair comparison with DeepDFT. In the invariant version of the DeepDFT model, edge features are represented solely by the distances between vertices, whereas the equivariant version incorporates both the distances and the directions of the edges as features.
>
> [1]. npj Computational Materials (2022) 8:183.
>
> > **Q2: What is ECD-HSE-id in Table 2? could authors clearly define or explain ECD-HSE-id in the paper, as this term is not explained.**
>
> Thank you for your feedback and for pointing out the issue regarding the undefined term ECD-HSE-id. We will ensure that this term is properly defined to enhance clarity and precision in the manuscript. We define ECD-HSE-id as follows: the dataset comprises 7,147 HSE-calculated data points, which are partitioned into 5,647 for training, 500 for validation, and 1,000 for testing. This task is designed to train a model solely on HSE data, serving as a baseline for comparison with a model pretrained on PBE data.
>
> > **Q3： Reviewer would suggest to authors to include a detailed description of structure of each dataset entry, possibly in appendix, if not already included.**
>
> We sincerely appreciate the reviewer’s constructive feedback. In response, we provided a dataset partition file, filelist.txt, in the source code, which lists the structure file names for each dataset split.
>
> > **Q4: Could authors provide more context about the Choudhary 2018 HSE-based dataset model?, including reasons for its performance characteristics, such as its error and time performance.**
>
> We sincerely thank the reviewer for their careful observation. We have provided the all experimental data points from the Choudhary 2018 HSE-based dataset model in the appendix of revised manuscript. These data were obtained through first-principles calculations using the HSE functional. Additionally, we have included a scatter plot comparing the calculated results with experimental values. It is worth noting KCl and CaO, typically exhibit larger errors. We also present the distribution of computation times for each data point. Materials with a higher number of atoms and elements located later in the periodic table, such as MoSe2 and ZrO2, exhibit the longest computation times.
>
>
> > **Q5: For Table 5, are there error bars? I am assuming the eV is band gap error, what is the acceptable error for band gap prediction?**
>
> We appreciate the reviewer’s thoughtful reminder. For Table 5, there are no error bars because the AI model uses a published pretrained model, and the predictions are deterministic for a given 3D material structure. Regarding acceptable error for band gap prediction, it depends on the application, but typically an error within 0.2–0.5 eV is considered acceptable for most materials design tasks.
>
> > **Q6： What is reason for discrepancy between optimal and achieved ratio in Table 3, and how was optimal ratio determined?**
>
> Thank you for the question. The discrepancy between the optimal and achieved ratios reflects inherent limitations in the accuracy of predicted charge density. This is primarily due to the dataset's wide atomic distribution (1~444 atoms), with medium-to-large systems (over 80 atoms) having higher degrees of freedom, requiring more DFT iterations (N) and making charge density prediction more challenging. The optimal ratio represents the minimum number of iterations, which is 1.

---

### Meta-Review · Area_Chair_eHzd · 2024-12-19

**Metareview:**

The paper introduces a repository for predicting Electronic Charge density prediction of crystalline materials, comprising of 140K crystal structures with medium precision PBE and high precision HSE data.

Strengths:
1.  This dataset offers a robust foundation for advancing scalable, high-accuracy electronic structure predictions in materials science. The paper conducts a comprehensive benchmarking of multiple models, assessing their performance across various tasks and metrics, including both in-distribution and out-of-distribution scenarios, which enhances the reliability and applicability of the findings [Reviewer 4VpH]

2.  Using the ChargE3Net model as a baseline, the authors evaluate its performance across multiple tasks, including prediction accuracy on the ECD dataset and the acceleration of DFT computations. Overall, this paper makes a valuable contribution to machine learning in quantum chemistry and computational materials science. This is a significant contribution to this area. Due to the computational cost to carefully converge the charge densities, it is not easy to generate or to find publicly available data sets. Thus, this work will clearly benefit the researchers in this area. [Reviewer NK1Z]

3. The dataset includes 140,646 stable crystal geometries with medium-precision PBE functional, and high accuracy HSE from subsets of the meticulously curated geometries. The computational efforts used in this study is immense, and the resulting benchmark could be a valuable asset to the ML community [Reviewer Tq9X]

Authors have patiently and objectively addressed questions and concerns raised by the reviewers during the rebuttal phase.

Weakness:
Most of the weaknesses highlighted by the reviewers were addressed by the authors during the rebuttal phase. However, from the results, the MAE of CrystalNet-TL  is close to ECD-PBE HSE (0.70 vs 0.65) but the required time is much lower than ECD-PBE HSE [Reviewer 4VpH]. Although this is a minor weakness.

**Additional Comments On Reviewer Discussion:**

Authors have patiently and objectively addressed questions and concerns raised by the reviewers during the rebuttal phase.

Reviewers Tq9X, NK1Z had closely engaged with the authors during rebuttal and have raised their scores, based on the clarifications.

---

### Decision · Program_Chairs · 2025-01-22

Accept (Oral)